# FOCAL: Contrastive Learning for Multimodal Time-Series Sensing Signals in Factorized Orthogonal Latent Space

**Shengzhong Liu**[*], **Tomoyoshi Kimura**[†], **Dongxin Liu**[‡], **Ruijie Wang**[†], **Jinyang Li**[†],
**Suhas Diggavi**[§], **Mani Srivastava**[§], **Tarek Abdelzaher**[†]
[*]Shanghai Jiao Tong University, [†]University of Illinois at Urbana-Champaign
[‡]Meta, [§]Univeristy of California, Los Angeles
shengzhong@sjtu.edu.cn, {tkimura4, ruijiew2, jinyang7, zaher}@illinois.edu
dxliu@meta.com, {suhas, mbs}@ee.ucla.edu

## Abstract

This paper proposes a novel contrastive learning framework, called FOCAL, for extracting comprehensive features from multimodal time-series sensing signals through self-supervised training. Existing multimodal contrastive frameworks mostly rely on the shared information between sensory modalities, but do not explicitly consider the exclusive modality information that could be critical to understanding the underlying sensing physics. Besides, contrastive frameworks for time series have not handled the temporal information locality appropriately. FOCAL solves these challenges by making the following contributions: First, given multimodal time series, it encodes each modality into a factorized latent space consisting of shared features and private features that are orthogonal to each other. The shared space emphasizes feature patterns consistent across sensory modalities through a modal-matching objective. In contrast, the private space extracts modality-exclusive information through a transformation-invariant objective. Second, we propose a temporal structural constraint for modality features, such that the average distance between temporally neighboring samples is no larger than that of temporally distant samples. Extensive evaluations are performed on four multimodal sensing datasets with two backbone encoders and two classifiers to demonstrate the superiority of FOCAL. It consistently outperforms the state-of-the-art baselines in downstream tasks with a clear margin, under different ratios of available labels. The code and self-collected dataset are available at https://github.com/tomoyoshki/focal.

## 1 Introduction

As a representative self-supervised learning (SSL) paradigm, contrastive learning (CL) has achieved unprecedented success in vision tasks [3, 14, 4, 12, 55] and are increasingly leveraged in learning from time series [10, 7, 63, 65, 36, 58]. However, many IoT applications [39, 56, 33] rely on heterogeneous sensory modalities to collaboratively perceive the physical surroundings and lead to a more complicated learning space. This paper aims to build a contrastive learning framework that maximally extracts complementary information from multimodal time-series sensing signals [23, 60],. The key challenge is to define appropriate similarity/distance measures (*i.e.*, who should be close to whom) in the joint multimodal embedding space that facilitates the downstream tasks [53, 57].

Existing contrastive frameworks for time series either ignore the heterogeneity among sensory modalities and are limited to instance-level discrimination [22, 17], or are designed to extract only shared information across sensory modalities [48, 6, 35]. For instance, as a representative framework, CMC [48] builds on the hypothesis that only information shared across views corresponds to the

37th Conference on Neural Information Processing Systems (NeurIPS 2023).

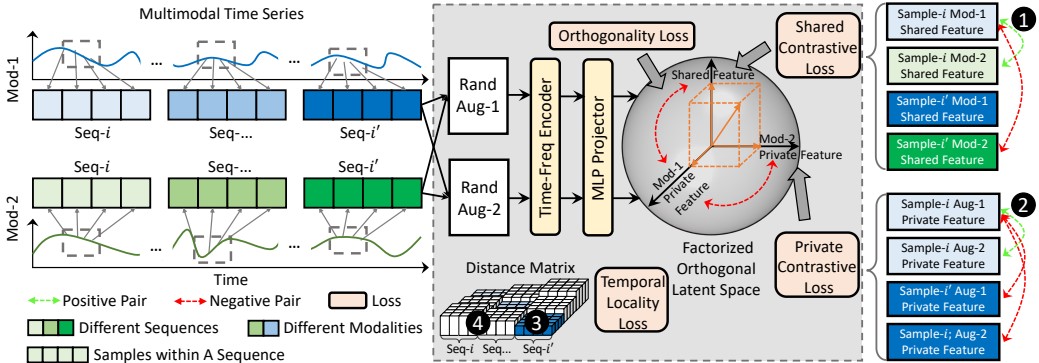

Figure 1: Overview of the FOCAL framework. Best viewed in color.

actual physical target. We argue that the strength of multimodal sensing lies in that the collaborating modalities not only share information but also enhance each other by providing complementary information exclusive to the modalities. The pretraining objective in contrastive learning should be calibrated to extract all semantically meaningful and generalizable patterns.

In addition, we observe that existing contrastive frameworks for time series do not handle the temporal information locality in a proper way. Temporal contrastive works (*e.g.*, TNC [49]) force the temporally close samples to be positive pairs of similarity 1 and force the temporally distant samples to be negative pairs of similarity 0, which may contradict long-term seasonality patterns. For example, a circling motorcycle may cause periodical vibration patterns to nearby microphone arrays, violating the above strict contrastive objective. Alternatively, TFC [65] optimizes the consistency between time-domain and frequency-domain representations, preventing neither encoder from simultaneously extracting features from the time-frequency spectrogram, which are known to achieve superior performance in learning from sensing signals [59].

To overcome these limitations, we propose a novel contrastive framework, FOCAL, for self-supervised representation learning from multimodal time-series sensing data. An overview is presented in Figure 1. Motivated by [43], we encode the input (*i.e.*, fixed-length window of sensor readings) of each modality into a factorized orthogonal latent space, composed of a shared space and a private space. The shared features and private features of the same modality, as well as the private features between different modalities, are mutually orthogonal to emphasize their semantical independence. The shared space is designed

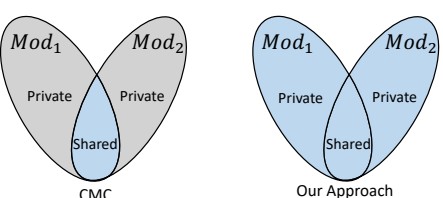

Figure 2: Information diagram between CMC and the proposed FOCAL. Figure adapted from [48]. Blue color denotes used information sectors.

to capture information consistency across modalities, while the private space captures modality-exclusive but transformation-invariant information. As we show in Figure 2, FOCAL outperforms CMC [48] by further emphasizing modality-exclusive sectors of target information. Besides, we define a temporal structural constraint to the distance measure of modality embeddings through a loose ranking constraint between temporally "close sample pairs" and "distant sample pairs". During the pertaining, we use a sampling strategy that randomly selects multiple short sequences with fixed length (*e.g.*, 4 samples per sequence) to constitute training batches. Instead of performing temporal contrastive tasks as [49], we confine the average intra-sequence distance between temporally neighboring samples (*i.e.*, ❸ in Figure 1) to be no larger than the average inter-sequence distances between temporally distant samples (*i.e.*, ❹ in Figure 1). It simultaneously addresses overall temporal information locality and tolerates occasional violations of locality caused by periodicity, and it turns out to accelerate the pretraining convergence in practice.

On the one hand, we justify the feature orthogonality constraints from two aspects. First, the shared features and private features of a modality should be orthogonal, such that the private space can avoid reusing the shared information but instead exploit modality-exclusive discriminative information. Second, the private features of different modalities should be orthogonal to each other. Otherwise, the overlapped semantics should be included in their shared space. In summary, the orthogonality

constraint is imposed to refine the heterogeneous modality-exclusive information that is undervalued in existing cross-modal contrastive tasks.

On the other hand, the temporal information locality within time series is defined through coarse-grained distance orders for two reasons. First, we do not always regard temporally close samples as positive pairs (with similarity 1), such that fine-grained differences between neighboring samples can be included. Second, we only enforce the temporal constraint at the statistical average scale by comparing the average intra-sequence distances to the average inter-sequence distances to fit potential exceptions caused by long-term seasonal signal patterns and significantly reduce the computational complexity because we avoid traversing the ranking losses within all sample triplets [15].

In summary, we define the following four perspectives of pretraining objectives in FOCAL.

- **Modality Consistency in Shared Space:** In the shared space, we push the features of different modalities of the same sample to be similar to each other compared to randomly mismatched modality features from two temporally distant samples.
- **Transformation Consistency in Private space:** In the private space, we push the modality features under two random augmentations to be similar to each other compared to modality features from two samples.
- **Orthogonality Constraint:** We enforce the shared feature and private feature of the same modality, as well as private features of different modalities, to be orthogonal to each other.
- **Temporal Locality Constraint:** We restrict the average distance of samples within a short time sequence to be no larger than the average distance between two random sequences.

We extensively evaluate FOCAL against eleven state-of-the-art baselines on four multimodal sensing datasets with two different backbone encoders (DeepSense [59] and Swin-Transformer [26]). The finetuning performance is evaluated with two light-weight classifiers (*i.e.*, linear and KNN classifier). It consistently achieves higher accuracy and F1 scores than the baselines. We also break down the contribution of individual components through step-by-step ablation studies.

## 2   Problem Formulation

Assume we have a set of $P$ sensory modalities $\mathcal{M} = \{M_1, M_2, \ldots, M_P\}$, and a large set of $N$ unlabeled pretraining data from all sensory modalities $\mathcal{X} = \{\mathbf{x}_1, \mathbf{x}_2, \ldots, \mathbf{x}_N\}$. Each sample is a fixed-length window of signals from all sensory modalities. For each sample $\mathbf{x}_i$, we use $\mathbf{x}_{ij}$ to denote the input from sensory modality $M_j$. The original input of each sensory modality is a multivariate time series, and we apply Short-Time Fourier Transform (STFT) as the preprocessing procedure to extract the sample time-frequency representation. As we introduce in Appendix B, the processed modality input is $\mathbf{x}_{ij} \in \mathbb{R}^{C \times I \times S}$, where $C$ denotes the number of input channels, $I$ denotes the number of time intervals within a sample window, and $S$ denotes the spectrum length after applying the Fourier transform to each interval[1]. We have a set of backbone encoders $\mathcal{E} = \{E_1, E_2, \ldots, E_P\}$ such that the encoder $E_i$ of modality $M_i$ can encode the modality input $\mathbf{x}_{ij}$ into an embedding vector $\mathbf{h}_{ij} = E_j(\mathbf{x}_{ij}) \in \mathbb{R}^K$, where $K$ is the unified dimension of modality embedding vectors. We use $\langle \mathbf{h}_{ij}, \mathbf{h}_{i'j'} \rangle$ to denote the inner product between two embedding vectors $\mathbf{h}_{ij}$ and $\mathbf{h}_{i'j'}$. The encoders extract both the time and frequency patterns from the preprocessed input. We also have a small set of $N'$ supervised samples for finetuning $\mathcal{X}^s = \{\mathbf{x}_1^s, \mathbf{x}_2^s, \ldots, \mathbf{x}_{N'}^s\}$, where each sample $\mathbf{x}_i^s$ is associated with a label $y_i^s$. Please note $\mathcal{X}^s$ is not necessarily a subset of the pretraining set $\mathcal{X}$ due to potential domain adaptations, *i.e.*, $\mathcal{X}^s \nsubseteq \mathcal{X}$. The objective of self-supervised pretraining is to use the unlabeled dataset $\mathcal{X}$ to optimize the parameters of modality encoders $\mathcal{E}$ such that the model accuracy finetuned on the supervised dataset $\mathcal{X}^s$ is maximized.

## 3   FOCAL Framework

In this section, we start with an overview of the framework and then separately introduce the two main functional components: (1) contrastive learning in factorized orthogonal latent space, and (2) temporal structural constraint.

---

[1]For notational simplicity, we do not differentiate the sampling rates of sensory modalities, but we can handle the case since each modality has a separate feature encoder.

## 3.1 Overview

The key questions to answer in designing a contrastive learning framework include the *selection of positive/negative pairs* and the *contrastive loss functions*. Compared to image input about which we have almost no knowledge without human annotations, the meta-level information of multimodal time-series input, *i.e.*, *cross-modal correspondence* and *temporal information locality*, can be effectively leveraged during the pretraining to shape the learned joint embedding space. As we show in Figure 1, FOCAL groups multiple randomly sampled fixed-length (*i.e.*, $L$ samples) short sequences of samples into a batch $\mathcal{B}$, with cardinality $|\mathcal{B}| = B$, and only conducts contrastive comparisons within the batch without any memory banks. Each modality input $\mathbf{x}_{ij}$ goes through two randomly selected augmentations to generate two augmented versions, and each augmented input is separately encoded by the modality encoder $E_j$ to get two versions of modality embeddings $\mathbf{h}_{ij}$ and $\tilde{\mathbf{h}}_{ij}$. The encoder network structures are not this paper's original contribution, so we leave them in Appendix D.

As the output of modality encoding, each modality embedding $\mathbf{h}_{ij}$ is projected through a non-linear multilayer perceptron (MLP) projector into two orthogonal embeddings, a shared embedding $\mathbf{h}_{ij}^{shared}$ and a private embedding $\mathbf{h}_{ij}^{private}$. The two embeddings share all encoder layers except the MLP projector. Separate contrastive learning tasks are applied to each embedding to capture different aspects of information. At the same time, shared-private and private-private modality features of the same sample are mutually orthogonal. In addition, we apply a temporal structural constraint between intra-sequence distances and inter-sequence distances to both projected modality embeddings.

## 3.2 Multimodal Contrastive Learning in Factorized Orthogonal Space

In multimodal collaborative sensing, information from different sensory modalities is not fully overlapped, thus extracting modality-exclusive discriminative information can reinforce the shared information during the downstream task finetuning. Projecting each modality embedding into separate shared space and private space while applying the orthogonality constraints avoids the shared information being reused in the private space optimization.

**Contrastive Task for Shared Space:** We use the shared feature space to learn modality consistency information. Specifically, we only consider samples within a batch $\mathcal{B}$ but across different short sequences, and assume the same random augmentation is applied. We iterate over all pairs of modalities $M_j$ and $M_{j'}$, regarding different modality embeddings of the same sample ($\mathbf{x}_{ij}^{shared}, \mathbf{x}_{ij'}^{shared}$) (*e.g.*, ❶ in Figure 1) as positive pairs, and regard two random modality embeddings from different samples ($\mathbf{x}_{ij}^{shared}, \mathbf{x}_{i'j'}^{shared}$) as negative pairs. We calculate the following InfoNCE loss [48],

$$\mathcal{L}_{shared} = -\sum_{i} \sum_{M_j, M_{j'} \in \mathcal{M}, j \neq j'} \log \frac{\exp\left(\langle \mathbf{h}_{ij}^{shared}, \mathbf{h}_{ij'}^{shared} \rangle / \tau \right)}{\sum_{i' \in \mathcal{B}} \exp\left(\langle \mathbf{h}_{ij}^{shared}, \mathbf{h}_{i'j'}^{shared} \rangle / \tau \right)}, \tag{1}$$

where $\tau$ is a temperature parameter that controls the penalties on hard negative samples [52].

**Contrastive Task for Private Space:** We use the private feature space to learn modality-exclusive information that is useful in discriminating different sequences within a batch $\mathcal{B}$, through capturing transformation consistency information. Specifically, for each modality $M_j$, we consider the encoded embeddings of two randomly augmented versions of the same samples as positive pairs (*e.g.*, ❷ in Figure 1), *i.e.*, ($\mathbf{h}_{ij}^{private}, \tilde{\mathbf{h}}_{ij}^{private}$), where we use $\tilde{\mathbf{h}}_{ij}^{private}$ to denote a differently augmented variant. The remaining 2B-2 modality embeddings in the batch are considered negative pairs to $\mathbf{h}_{ij}^{private}$. We use the NT-Xent [3] loss for the private space contrastive task,

$$\mathcal{L}_{private} = -\sum_{i} \sum_{M_j \in \mathcal{M}} \log \frac{\exp\left(\langle \mathbf{h}_{ij}^{private}, \tilde{\mathbf{h}}_{ij}^{private} \rangle / \tau \right)}{\sum_{i' \in \mathcal{B}, i' \neq i} \exp\left(\langle \mathbf{h}_{ij}^{private}, \mathbf{h}_{i'j}^{private} \rangle / \tau \right) + \sum_{i' \in \mathcal{B}} \exp\left(\langle \mathbf{h}_{ij}^{private}, \tilde{\mathbf{h}}_{i'j}^{private} \rangle / \tau \right)}. \tag{2}$$

**Orthogonality Constraint:** To enforce the orthogonality constraint between the shared feature and private feature of the same modality, as well as the private features between different modalities, such that they can capture independent semantic information in the factorized space, we apply a cosine

embedding loss that directly minimizes their angular similarities,

$$\mathcal{L}_{orthogonal} = \sum_i \sum_{M_j \in \mathcal{M}} \langle \mathbf{h}_{ij}^{shared}, \mathbf{h}_{ij}^{private} \rangle + \sum_i \sum_{M_j, M_j' \in \mathcal{M}, j \neq j'} \langle \mathbf{h}_{ij}^{private}, \mathbf{h}_{ij'}^{private} \rangle. \quad (3)$$

### 3.3 Temporal Structural Constraint

Appropriately shaping the temporal information locality in latent modality embedding space is challenging. First, simply considering temporally close samples as positive pairs and pushing their semantical similarities to 1 as [63, 49] can be problematic because they ignore the continuously evolving factors (*e.g.*, distance) that make differences between temporally neighboring samples. Second, it is also impractical to predefine a fixed similarity curve with respect to the time difference between two samples, which is highly context-dependent and can not be known in advance.

Considering the complexity of temporal information correlations, we stop defining positive versus negative pairs in the temporal dimension. Instead, we only apply a loose ranking loss to specify the relationships of distances at the coarse-grained sequence level (intra-sequence distance vs. inter-sequence distance) as an information regularization [9]. Given a sequence, we restrict the average distances between samples within the sequence (*e.g.*, ❸ in Figure 1) to be no larger than their average distance to samples from other sequences (*e.g.*, ❹ in Figure 1). To accommodate occasional violations of temporal locality caused by long time periodicity (such that temporally distant samples could be more similar than neighboring samples), and further reduce the computational complexity caused by multiplicative traverse of sample triplets, we restrict the average intra-sequence distances to be no larger than average inter-sequence distances.

We calculate the distance in the Euclidean space, to facilitate the downstream Euclidean classifiers. Given the sample-level distance matrix $\mathbf{D} \in \mathbb{R}^{BL \times BL}$, we first compute the sequence-level mean distance matrix $\bar{\mathbf{D}} \in \mathbb{R}^{B \times B}$ through aggregating sample-level distances[2], such that the average distance between sequence $s$ and sequence $s'$ is $\bar{D}_{ss'} = 1/L^2 \sum_{i \in s, i' \in s'} D_{pq}$, then the temporal locality loss is defined by,

$$\mathcal{L}_{temporal} = \sum_s \sum_{s' \neq s} \max(\bar{D}_{ss} - \bar{D}_{ss'} + \text{margin}, 0), \quad (4)$$

where the margin is the predefined degree of separation. It is worth noting that the temporal structural constraint is applied to holistic modality embeddings, including both the shared and private parts.

### 3.4 Overall Training Objective

In summary, FOCAL pretraining simultaneously considers latent correlations between (1) synchronized embeddings of different modalities; (2) differently augmented modality embeddings; and (3) temporally neighboring sample embeddings. It minimizes the following overall loss,

$$\mathcal{L} = \mathcal{L}_{shared} + \lambda_p \cdot \mathcal{L}_{private} + \lambda_o \cdot \mathcal{L}_{orthogonal} + \lambda_t \cdot \mathcal{L}_{temporal}, \quad (5)$$

where $\lambda_p$, $\lambda_o$, and $\lambda_t$ are hyperparameters that control the weights of each loss component.

## 4 Evaluation

In this section, we evaluate FOCAL by comparing it with 11 popular SOTA self-supervised learning frameworks on four different datasets. We start with the experimental setups and then introduce the main evaluation results, followed by comprehensive ablation studies.

### 4.1 Experimental Setup

**Datasets:** (1) **MOD** is a self-collected dataset using acoustic (8000Hz) and seismic (100Hz) signals to classify moving vehicle types. It includes 6 different vehicle types and 1 class of human walking. (2) **ACIDS** is an ideal dataset for vehicle classification using acoustic signals and seismic signals

---

[2]The sample-level diagonal elements that always have distance 0 are skipped during the aggregation.

Table 1: Statistical Summaries of Evaluated Datasets.

| Dataset | Classes | Modalities (Freq) | Sample Length | Interval (Overlap) | #Samples | #Labels |
|---------|---------|-------------------|---------------|--------------------|----------|---------|
| MOD | 7 | acoustic (8000Hz), seismic (100Hz) | 2 sec | 0.2 sec (0%) | 39,609 | 7,335 |
| ACIDS | 9 | acoustic, seismic (both 1025Hz) | 1 sec | 0.25 sec (50%) | 27,597 | 27,597 |
| RealWorld-HAR | 8 | acc, gyro, mag, lig (all 50Hz) | 5 sec | 1 sec (50%) | 12,887 | 12,887 |
| PAMAP2 | 18 | acc, gyr, mag (all 100Hz) | 2 sec | 0.4 sec (50%) | 9,611 | 9,611 |

Table 2: Finetune Results with Linear Classifier

| Dataset | | MOD | | ACIDS | | RealWorld-HAR | | PAMAP2 | |
|---------|---------|-----|-----|-------|-----|---------------|-----|--------|-----|
| Encoder | Framework | Acc | F1 | Acc | F1 | Acc | F1 | Acc | F1 |
| | Supervised | 0.9404 | 0.9399 | **0.9566** | 0.8407 | 0.9348 | **0.9388** | **0.8849** | **0.8761** |
| DeepSense | SimCLR | 0.8855 | 0.8855 | 0.7438 | 0.6101 | 0.7138 | 0.6841 | 0.6802 | 0.6583 |
| | MoCo | 0.8808 | 0.8812 | 0.7717 | 0.6205 | 0.7859 | 0.7708 | 0.7559 | 0.7387 |
| | CMC | 0.9196 | 0.9186 | 0.8443 | 0.7244 | 0.7975 | 0.8116 | 0.7906 | 0.7706 |
| | MAE | 0.5981 | 0.5993 | 0.6644 | 0.5618 | 0.7565 | 0.7515 | 0.7114 | 0.6158 |
| | Cosmo | 0.8989 | 0.8998 | 0.8511 | 0.6929 | 0.8956 | 0.8888 | 0.8356 | 0.8135 |
| | Cocoa | 0.8774 | 0.8764 | 0.6644 | 0.5359 | 0.8465 | 0.8488 | 0.7603 | 0.7187 |
| | MTSS | 0.4153 | 0.3582 | 0.4352 | 0.2441 | 0.2989 | 0.1405 | 0.3541 | 0.1795 |
| | TS2Vec | 0.7669 | 0.7648 | 0.5224 | 0.3587 | 0.6595 | 0.5984 | 0.5729 | 0.4715 |
| | GMC | 0.9257 | 0.9267 | 0.9096 | 0.7929 | 0.8869 | 0.8948 | 0.8119 | 0.7860 |
| | TNC | 0.9518 | 0.9528 | 0.8237 | 0.6936 | 0.8892 | 0.8971 | 0.8387 | 0.8143 |
| | TS-TCC | 0.8707 | 0.8735 | 0.7667 | 0.6164 | 0.8073 | 0.8010 | 0.7776 | 0.7250 |
| | FOCAL | **0.9732** | **0.9729** | 0.9516 | **0.8580** | **0.9382** | 0.9290 | 0.8588 | 0.8463 |
| | Supervised | 0.8948 | 0.8931 | 0.9137 | 0.7770 | 0.9313 | 0.9278 | **0.8612** | 0.8384 |
| SW-T | SimCLR | 0.9250 | 0.9247 | 0.9128 | 0.8144 | 0.7046 | 0.7220 | 0.7705 | 0.7424 |
| | MoCo | 0.9390 | 0.9384 | 0.9174 | 0.8100 | 0.7813 | 0.8024 | 0.7717 | 0.7313 |
| | CMC | 0.9129 | 0.9105 | 0.8128 | 0.6857 | 0.8840 | 0.8955 | 0.8080 | 0.7901 |
| | MAE | 0.7803 | 0.7772 | 0.8516 | 0.7023 | 0.8829 | 0.8813 | 0.7910 | 0.7606 |
| | Cosmo | 0.3429 | 0.3378 | 0.7110 | 0.6086 | 0.8604 | 0.8169 | 0.7741 | 0.7366 |
| | Cocoa | 0.7040 | 0.7038 | 0.7096 | 0.5794 | 0.8892 | 0.8861 | 0.7689 | 0.7317 |
| | MTSS | 0.4206 | 0.4163 | 0.3429 | 0.2250 | 0.5136 | 0.4370 | 0.2847 | 0.1714 |
| | TS2Vec | 0.7254 | 0.7174 | 0.7183 | 0.5748 | 0.6151 | 0.5955 | 0.6195 | 0.5426 |
| | GMC | 0.8640 | 0.8611 | 0.9402 | 0.7766 | 0.9319 | 0.9379 | 0.8312 | 0.8083 |
| | TNC | 0.8533 | 0.8539 | 0.8352 | 0.7372 | 0.8817 | 0.8784 | 0.8013 | 0.7506 |
| | TS-TCC | 0.8734 | 0.8735 | 0.9041 | 0.7547 | 0.8731 | 0.8454 | 0.7997 | 0.7260 |
| | FOCAL | **0.9805** | **0.9800** | **0.9489** | **0.8262** | **0.9451** | **0.9503** | 0.8580 | **0.8401** |

(both in 1025Hz). It includes data on 9 types of ground vehicles in 3 different terrains. (3) **RealWorld-HAR [46]** is a public dataset using accelerometer, gyroscope, magnetometer, and light signals (all in 50Hz) to recognize 8 common human activities. (4) **PAMAP2 [41]** is another public dataset using accelerometer, gyroscope, and magnetometer signals (all in 100Hz) to recognize 18 different physical activities. The length of the samples and the intervals, as well as the time overlap ratios between intervals within samples of each dataset, as listed in Table 1, are configured to achieve the best-supervised classification performance.

**Data Augmentations:** Candidate augmentations are defined in both the time domain before STFT and the frequency domain after STFT, where only one out of them is randomly selected in each forward pass. Time-domain augmentations include scaling, permutation, negation, time warp, magnitude warp, horizontal flip, jitter, channel shuffle, and time masking; frequency-domain augmentations include phase shift and frequency masking. Details can be found in Appendix B.

**Baselines:** We consider 12 baselines in total, including a supervised benchmark, three representative self-supervised learning frameworks from vision tasks that perform instance discrimination (SimCLR [3], MoCoV3 [5], and MAE [13]), four modality-matching contrastive frameworks for multimodal input (CMC [48], Cosmo [35], Cocoa [6], GMC [37]), three SOTA contrastive frameworks for time series (TS2Vec [63], TNC [49], TS-TCC [8]), and one predictive baseline (MTSS [42]). Their detailed introductions can be found in Appendix C. All compared frameworks use the same encoder structures (except for minor differences in module orders). The evaluation metrics are accuracy and (macro) F1 score.

**Backbone Encoders:** We apply two different modality encoders in this paper. (1) **DeepSense** [59] uses convolutional layers to extract localized feature patterns within each time interval, and then uses recurrent layers to aggregate information across all intervals within a sample window. (2) **Swin-Transformer (SW-T)** [26] uses stacked Transformer blocks to extract local information from shifted windows in a hierarchical manner. Their details and configurations can be found in Appendix D.

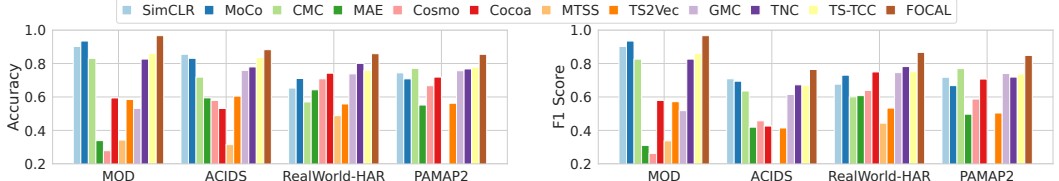

Figure 4: Finetuning results with KNN classifiers (K=5), with SW-T encoder.

Both models are implemented in PyTorch 1.14 and optimized with an AdamW [28] optimizer under cosine learning rate schedules [27]. Without special specifications, we report the results on SW-T backbone. Detailed training configurations are introduced in Appendix E.

**Finetune Classifiers:** We evaluate with two lightweight classifiers during the finetune stage. **Linear probling** adds a linear layer on top of the pretrained sample features (or concatenated modality features) with the encoder fixed during fine-tuning. **KNN classifier** directly uses the voting of 5 nearest neighbors (based on the Euclidean distances between samples) in the finetuning training samples to generate the predicted labels for the testing samples.

## 4.2 Finetune Results

**Linear Probing:** Table 2 presents the finetune results using a linear classifier (The complete results can be found in Appendix F). FOCAL consistently achieves the best accuracy and F1 score across different datasets and backbone encoders. It outperforms CMC by 4.48% to 18.01% in accuracy, demonstrating the importance of extracting private modality information in addition to the shared information. SimCLR and MoCo achieved suboptimal performance on HAR datasets (*i.e.*, RealWorld-HAR and PAMAP2) which have more than two modalities, showing that instance discrimination alone as the pretext task can not learn comprehensive feature patterns from multiple modalities. GMC, as the SOTA framework for multi-modal time series, beats the contrastive frameworks for single-modal time series (TS2Vec, TNC, and TS-TCC) in most cases, but is secondary to FOCAL. On MOD dataset, FOCAL achieves higher accuracy and F1 score than the supervised model by taking advantage of massive unlabeled samples.

Besides, we plot the finetune results on the MOD dataset under three different label ratios (100%, 10%, and 1%) in Figure 3. FOCAL overall achieves better label efficiency during the finetuning. It attains 3.51% relative improvement with 100% labels and 10.56% relative improvement with 1% labels, compared to the best-performing baseline. Among the baselines, TNC performs best with 1% label but does not generalize well when more labels are available (*e.g.*, 100%).

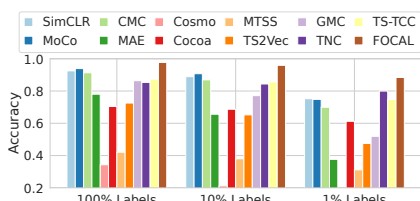

Figure 3: Linear probing under different label ratios on MOD with SW-T encoder.

**KNN Evaluation:** The evaluation results with a KNN classifier (K=5) using all available labels[3] are plotted in Figure 4 (See Appendix F for complete results). KNN operates in a non-parametric way and can be expressive if the learned latent space aligns well with the underlying semantics. For multi-modal contrastive frameworks (*i.e.*, FOCAL, CMC, Cosmo, Cocoa, GMC), we concatenate the modality features as the sample feature. FOCAL consistently performs better than the baselines across all datasets, proving its representational power in the non-parametric classification setting.

## 4.3 Additional Downstream Tasks

**Clustering:** In multi-modal learning, the clusterability of the learned modality representations is preferable. Meaningful representation clusters should be coherent with the underlying semantic labels. For each modality, we first use K-means clustering with pretrained modality embeddings to get the cluster labels and measure the consistency between the cluster labels and the ground-truth category labels, with two common clustering metrics adjusted rand score (ARI) [45] and normalized mutual information score (NMI) [1]. The results for five multi-modal frameworks are presented in Figure 5.

---

[3]In MOD, 7,335 samples out of the pretraining set are labeled. All other datasets are fully labeled.

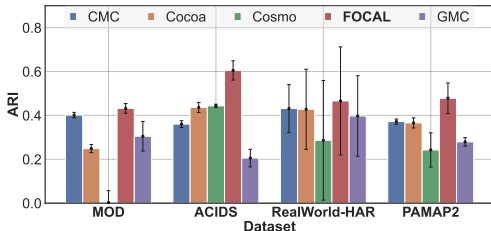 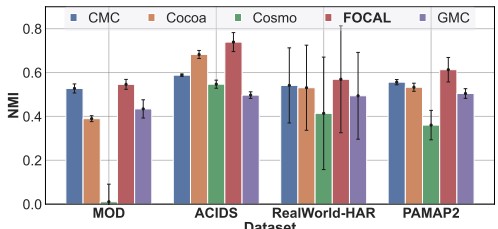

Figure 5: Clustering evaluation results. We use SW-T as the backbone encoder. The mean and standard deviation among sensory modalities are reported. Higher scores are better.

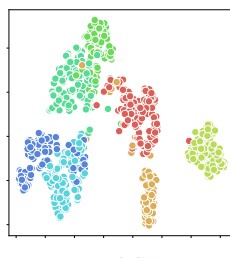 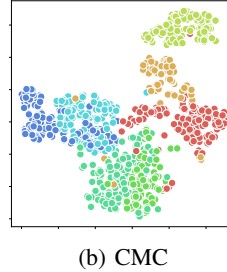 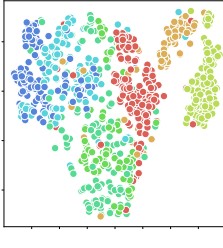 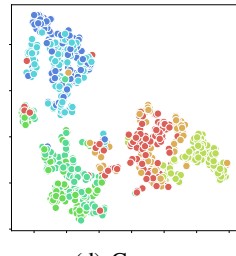

(a) FOCAL        (b) CMC        (c) GMC        (d) Cocoa

Figure 6: t-SNE visualization of the concatenated modality features (DeepSense encoder, MOD dataset). Different colors represent different object classes.

Table 3: Ablation Results with SW-T Encoder and Linear Classifier.

| Metrics | MOD | | ACIDS | | RealWorld-HAR | | PAMAP2 | |
|---|---|---|---|---|---|---|---|---|
| | Acc | F1 | Acc | F1 | Acc | F1 | Acc | F1 |
| FOCAL-noPrivate | 0.9296 | 0.9284 | 0.7981 | 0.7100 | 0.8869 | 0.8768 | 0.7938 | 0.7787 |
| FOCAL-noOrth | 0.9705 | 0.9692 | 0.9311 | 0.8261 | 0.9186 | 0.9257 | 0.8371 | 0.8233 |
| FOCAL-wDistInd | 0.5773 | 0.5502 | 0.4926 | 0.4157 | 0.9099 | 0.9084 | 0.6518 | 0.5503 |
| FOCAL-noTemp | 0.9671 | 0.9659 | 0.9456 | 0.8014 | 0.9361 | 0.9425 | 0.8367 | 0.8255 |
| FOCAL-wTempCon | 0.9363 | 0.9359 | 0.9287 | 0.7587 | 0.8793 | 0.8842 | 0.8391 | 0.8242 |
| FOCAL | **0.9805** | **0.9800** | **0.9489** | **0.8262** | **0.9451** | **0.9503** | **0.8580** | **0.8401** |

We separately evaluate the clusterability of each modality and report the mean and standard deviation among modalities. FOCAL is superior in learning representations better aligned with the target labels. Although CMC performs closely to FOCAL on MOD and RealWorld-HAR datasets, it performs poorly on ACIDS and PAMAP2 datasets. Besides, we qualitatively visualize the concatenated sample embeddings of the multi-modal contrastive frameworks after t-SNE [51] dimension reduction on MOD dataset in Figure 6. We can see that FOCAL achieved better separation among different classes than the compared baselines.

**Distance and Speed Classification:** In the MOD dataset, we separately collect new data samples recording the distance and speed of the vehicles. Data in this experiment is collected in a different environment with different vehicles from the pretraining data, thus accounting for potential domain shifts. The same pretrained models in previous experiments are used, and the results are summarized in Figure 7. The instance discrimination frameworks (*i.e.*, SimCLR and MoCo) are relatively more resilient than the modality

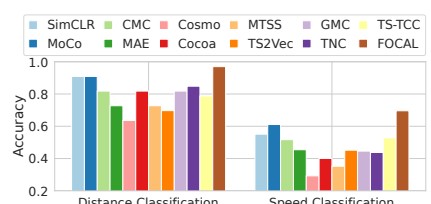

Figure 7: Additional downstream tasks on MOD dataset.

matching frameworks (*i.e.*, CMC, Cocoa, Cosmo, and GMC), while FOCAL fills this gap by capturing the transformation consistency information in private modality spaces.

## 4.4 Ablation Studies

Table 4: Benefits of Temporal Constraints to SOTA baselines on ACIDS

| Metrics | SimCLR | | MoCo | | CMC | | Cocoa | | GMC | |
|---|---|---|---|---|---|---|---|---|---|---|
| | Acc | F1 | Acc | F1 | Acc | F1 | Acc | F1 | Acc | F1 |
| wTemp | **0.7461** | **0.6938** | **0.7836** | **0.6618** | **0.8690** | 0.7090 | **0.8543** | **0.7665** | **0.9347** | **0.8109** |
| Vanilla | 0.7438 | 0.6101 | 0.7717 | 0.6205 | 0.8443 | **0.7244** | 0.6644 | 0.5359 | 0.9096 | 0.7929 |

Table 5: Benefits of Temporal Constraints to SOTA baselines on PAMAP2

| Metrics | SimCLR | | MoCo | | CMC | | Cocoa | | GMC | |
|---|---|---|---|---|---|---|---|---|---|---|
| | Acc | F1 | Acc | F1 | Acc | F1 | Acc | F1 | Acc | F1 |
| wTemp | **0.7129** | **0.6884** | **0.7800** | **0.7602** | 0.7804 | 0.7583 | **0.8442** | **0.8146** | **0.8253** | **0.8114** |
| Vanilla | 0.6802 | 0.6583 | 0.7559 | 0.7387 | **0.7906** | **0.7706** | 0.7603 | 0.7187 | 0.8119 | 0.7860 |

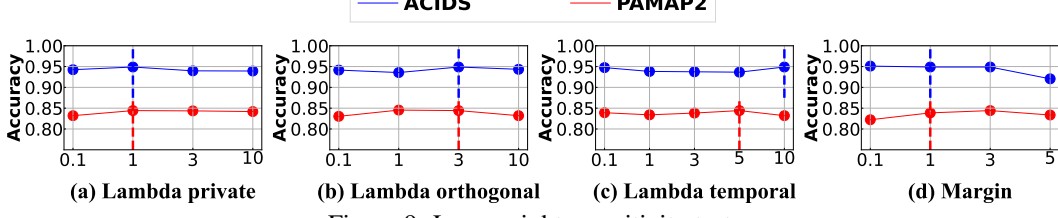

Figure 9: Loss weights sensitivity test.

**(a) Lambda private**  **(b) Lambda orthogonal**  **(c) Lambda temporal**  **(d) Margin**

**Setup:** Here we compare with five variants of FOCAL, including **FOCAL-noPrivate** (no private space), **FOCAL-noOrth** (no orthogonality constraint), **FOCAL-wDistInd** (replace orthogonality with distributional independence), **FOCAL-noTemp** (no temporal structural constraint), **FOCAL-wTempCon** (replace temporal structural constraint with the temporal contrastive task).

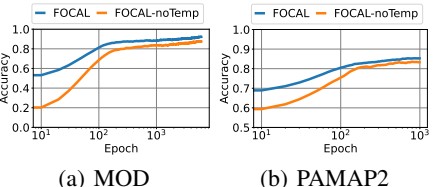

(a) MOD    (b) PAMAP2

Figure 8: Convergence curves.

**Analysis:** The ablation study results are summarized in Table 3. First, the accuracy decreases by 5.20%-5.90% after removing the private space and the related contrastive task, since only the shared information among modalities is leveraged. Second, on top of the private space task, both the orthogonality constraint and the temporal structural constraint further improve the accuracy by 1.39%-2.99%, which proves they both contribute positively to FOCAL. Third, replacing the geometrical orthogonality constraint with the distributional independence constraint causes significant degradation and may even cause the model training to collapse. Similarly, in our experiments, conducting temporal contrastive tasks leads to a noticeable accuracy decrease in three out of four datasets. To further demonstrate the contribution of our temporal structural constraint in accelerating the convergence during the pretraining, we use KNN classifier to periodically evaluate the quality of the learned representations. Concatenated modality embeddings are used as sample representations. We visualize the achieved KNN accuracy curves on MOD and PAMAP2 with and without the temporal constraint in Figure 8. The temporal constraint clearly improves the semantical structure of the learned embeddings in early epochs of pretraining, by rejecting obviously counter-intuitive parameter values that violate the constraint.

### 4.5 General Applicability of the Temporal Constraint

To validate the general applicability of the proposed temporal constraint, we apply it to multiple contrastive learning baselines (*i.e.*, SimCLR, MoCo, CMC, Cocoa, and GMC). Table 4 and 5 summarize the results on ACIDS and PAMAP2 respectively, and we have observed noticeable performance improvement in most cases (up to 18.99% on ACIDS and up to 8.39% on PAMAP2). It demonstrates that the temporal constraint can be used as a plugin to enhance existing contrastive learning frameworks for time-series data.

### 4.6 Sensitivity Test on Loss Weights

We also perform a sensitivity test on the loss weight values and plot the performance of FOCAL against different hyperparameters in Figure 9. We observe that FOCAL is generally robust against the

hyperparameter selections, with less than 2% accuracy fluctuations in all cases. For this reason, we did not perform a comprehensive hyperparameter search in our experiment. Besides, combining our observations in the ablation study that the private space task, orthogonality constraint, and temporal constraint all contribute positively to the performance of FOCAL, we conclude that the competition between learning objectives does not happen in FOCAL.

## 5    Related Works

**Self-Supervised learning for multimodal data.** Self-supervised learning from multimodal data has been extensively studied in the literature, including contrastive learning [48, 35, 6, 37, 38, 25, 32, 2], masked autoencoders (MAE) [13, 11, 19], and variational autoencoder (VAE) [18, 50, 21], surpassing conventional generation-based semi-supervised learning approaches [61]. As a leading paradigm in learning transferable and discriminative features [66, 24, 57] with a variety of successful applications [16, 29, 40, 30, 47], we mainly consider contrastive learning in this paper. On one hand, as we show in the experimental results, normal contrastive frameworks based on instance discriminations (SimCLR [20], MoCO [5], BYOL [12], SimSiam [4]) may lead to suboptimal results without accommodating the multimodal properties. On the other hand, existing contrastive learning frameworks for multimodal data [38, 48, 35, 6, 37] mostly focus on the consistency between modalities but ignore the information heterogeneity when they are collaboratively utilized in sensing tasks. CLIP [38], as a representative multimodal contrastive framework, mainly addresses the importance of using natural language to augment the visual models through learning their pairings. Similarly, both CMC [48] and GMC [37] highlight the information matching between modality embedding to another modality embedding or the joint embedding. Instead, in FOCAL, we simultaneously consider modality consistency and discriminative modality-exclusive information by designing corresponding contrastive tasks and imposing the information independence constraint. Besides, most existing multimodal contrastive learning frameworks are limited to vision and language modalities [62, 22, 47, 64, 31], while the domain knowledge might not be directly applicable to sensory modalities with time-series signals in IoT applications.

**Contrastive learning for time series.** There has been increasing interest in developing contrastive learning frameworks for time-series data [10, 63, 44, 7, 49, 22, 65, 36]. TS2Vec [63], TFC [65], TNC [49], and TS-TCC [7] were based on the time-series properties but did not consider the multimodal collaboration properties. Besides, TFC [65] and RF-URL [44] were designed from the consistency between time and frequency representations, or different time-frequency representations, while restricting the backbone encoder structures. Cosmo [35] and Cocoa [6] are the most recent attempts at contrastive learning from multi-modal time series, but they were not able to sufficiently utilize the complementary information from sensory modalities. CoST [54] proposed to separate the seasonal-trend representations for time series forecasting. Different from the existing works, the objective of FOCAL is to maximally extract complementary and discriminative features from multimodal sensory modalities, to facilitate the downstream recognition tasks. CLUDA [36] worked on the unsupervised domain adaptation (UDA) problem for time series, which is not directly comparable to our solution. In FOCAL, no expert features are assumed to be available as in ExpCLR [34].

## 6    Conclusion

We proposed a novel contrastive learning framework, FOCAL, for self-supervised learning from multimodal time-series sensing signals. FOCAL encodes each modality input into a factorized orthogonal latent space including shared features and private features. We learn each part by applying different contrastive learning objectives addressing the modality consistency in the shared space and the transformation invariance in the private space. Besides, we design a lightweight temporal structural constraint as an information regularization during the pretraining. Extensive evaluations on four multimodal sensing datasets with two encoder networks and two lightweight classifiers, demonstrate the superiority of FOCAL over 11 SOTA baselines, under different levels of supervision. Our future work would focus on extracting domain-invariant features in multi-vantage sensing applications to make the pretrained model resilient against task-unrelated environmental factors (*e.g.*, terrain, wind, sensor-facing directions).

## Acknowledgments and Disclosure of Funding

Research reported in this paper was sponsored in part by the Army Research Laboratory under Cooperative Agreement W911NF-17-20196, NSF CNS 20-38817, National Natural Science Foundation of China (No. 62332014), DARPA award HR001121C0165, DARPA award HR00112290105, DoD Basic Research Office award HQ00342110002, and the Boeing Company. Shengzhong Liu is also sponsored by the "Double-First Class" Startup Research Funding of Shanghai Jiao Tong University. The views and conclusions contained in this document are those of the author(s) and should not be interpreted as representing the official policies of the CCDC Army Research Laboratory, or the US government. The US government is authorized to reproduce and distribute reprints for government purposes notwithstanding any copyright notation hereon.

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

## Appendix

The appendix of this paper is structured as follows.

- Section A introduces the evaluated datasets and their statistics.
- Section B introduces the preprocessing procedure in our experiments.
- Section C details the compared baselines in our experiments.
- Section D introduces the used backbone model structure and their related configurations.
- Section E summarizes the training strategies we applied in this paper.
- Section F presents more comprehensive evaluation results.
- Section G lists the existing limitations of the work and potential solutions for future extensions.

## A   Datasets

The basic statistics for each dataset are summarized in Table 1.

**Moving Object Detection (MOD):** This is a self-collected dataset using sensor nodes consisting of a RaspberryShake 4D (from `https://raspberryshake.org/`) and a microphone array to collect the vibration signals caused by nearby moving vehicles. The data was collected from two different sites, where one was a former State park repurposed for research purposes, while the other was a large college parking lot. The RaspberryShake featured a geophone designed to measure seismic vibrations due to remote earthquakes. It was found to be much more sensitive to vibrations introduced by nearby moving objects than, say, accelerometers on a smartphone. In this dataset, we introduced each of seven different targets alternately in the vicinity of the sensor nodes: A Polaris off-road vehicle (from `https://ranger.polaris.com/`), a Chevrolet Silverado, a Warthog all-terrain Unmanned Ground Vehicle (from `https://clearpathrobotics.com/`), a Motorcycle, a Tesla, a Mustang, and a dismount human. Each target moved around at a different speed, while our sensors collected the corresponding seismic and acoustic signals. Only one target is considered during our experiments. The sampling rate for the seismic signal was 100Hz and the acoustic signal was collected under 16000Hz (which was downsampled to 8000Hz in the preprocessing). For each target, the collection lasted between 40 minutes to 1 hour. The training, validation, and testing datasets are randomly partitioned with a ratio of 8:1:1 at the sample level. (See IRB note.[4]) We do plan to release this dataset for public usage after the paper anonymization period.

**Acoustic-seismic identification Data Set (ACIDS):** ACIDS is an ideal dataset for developing and training acoustic/seismic classification/ID algorithms. The data was collected by 2 co-located acoustic/seismic sensor systems. There are over 270 data runs (single target only) from 9 different types of ground vehicles in 3 different environmental conditions. The ground vehicles were traveling at constant speeds from one direction toward the sensor systems passing the closest point of approach (CPA) and then away from the sensor systems. The microphone data is low-pass filtered at 400 Hz via a 6th-order filter to prevent spectral aliasing and high-pass filtered at 25 Hz via a 1st-order filter to reduce wind noise. The data is digitized by a 16-bit A/D at the rate of 1025 Hz. The CPA to the sensor systems varied from 25m to 100m. The speed varied from 5km/hr to 40km/hr depending upon the particular run, the vehicle, and the environmental condition. We randomly partition the runs into training, validation, and testing datasets with a ratio of 8:1:1. It is more challenging than MOD since domain shift caused by vehicle speed, distance, or terrain between training and testing can be included. No information related to the target types is revealed except the numerical labels.

**RealWorld-HAR [18]:** This is a public dataset using the accelerometer, gyroscope, magnetometer, and light signals to recognize 8 common human activities (climbing stairs down and up, jumping, lying, standing, sitting, running/jogging, and walking) from 15 subjects. Only the data collected from "waist" is used in our experiments. The sampling rate of all selected sensors is 100Hz. We use the

---

[4]The work was deemed Not Human Subjects Research (NHSR) because the purpose of the experiment was to test the performance of an AI algorithm in the presence of noise, as opposed to collecting data about humans. The humans who assisted with the experiment, in essence, acted as "lab technicians" who operate machinery for experimental purposes.

leave-one-out evaluation strategy where 10 random subjects are used for training, 2 subjects are used for validation, and 3 subjects are used for testing.

**Physical Activity Monitoring dataset (PAMAP2) [16]**: This dataset contains data of 18 different physical activities (e.g., walking, cycling, playing soccer, etc) performed by 9 subjects using inertial measurement units (IMUs) that are put at the chest, wrist (of dominant arm), and dominant side's ankle respectively. Only data collected from the "wrist" is used in our experiment. Each IMU records readings from a 3-axis accelerometer, gyroscope, and magnetometer. The sampling rates of all sensors are 100Hz. We use the leave-one-out evaluation strategy where 7 random subjects are used for training, and 2 subjects are used for testing.

# B  Data Preprocessing

In our data preprocessing, we first divide the time-series data into equal-length data samples and further segment each sample into overlapped/non-overlapped intervals. The signals within each interval are processed by the Fourier transform to obtain the spectrum. In this way, both the time-domain information and frequency-domain patterns are preserved. The generated time-frequency spectrogram is further fed into the backbone feature encoders. We define a set of data augmentations in both the time domain before the Fourier transform and the frequency domain after the Fourier transform. For each sample, only one random augmentation from either the time domain or the frequency domain is selected and applied. To further increase the randomness of data augmentations in multimodal applications, we let each modality have a probability of 0.5 to be processed by the selected random augmentation.

## B.1  Data Augmentations

We follow the common practices in [22, 7, 10, 19] to define the augmentations used in the time domain and frequency domain respectively.

### B.1.1  Time-Domain Augmentations

Here we list the used time-domain augmentations.

- **Scaling:** We multiply the input signals with values sampled from a Gaussian distribution.
- **Permutation:** Given intervals within a sample, we randomly permute the order of the intervals.
- **Negation:** The signal values are multiplied by a factor of -1.
- **Time Warp:** Randomly stretching/distorting the time locations of the signal values based on a smooth random curve.
- **Magnitude Warp:** The magnitude of each time series is multiplied by a curve created by cubicspline with a set number of knots at random magnitudes.
- **Horizontal Flip:** The entire time series of the sample is flipped in the time direction.
- **Jitter:** We add random Gaussian noise to signals.
- **Channel Shuffle:** We randomly shuffle the channels of multi-variate time-series data (*e.g.*, X, Y, Z dimensions of three-axis accelerometer input).
- **Time Masking:** We randomly mask a portion of the time intervals within a sample window with 0.

### B.1.2  Frequency-Domain Augmentations

Here we list the used frequency-domain augmentations.

- **Phase Shift:** Given the complex frequency spectrum, we add a random value between $-\pi$ to $\pi$ to their phase values.
- **Frequency Masking:** We randomly mask a portion of frequency ranges with 0.

## C  Baselines

**Supervised:** We train the whole model including the encoder and linear classifier in a fully supervised manner using all available labels.

**SimCLR** [1] is a simple yet powerful contrastive learning framework proposed for vision tasks. For this work, we randomly formulate batches. During pretraining, we apply random augmentations to generate two different views of each sample, with a contrastive objective of bringing different transformations (augmentations) of the same samples closer while repelling the representations of different samples. The framework optimizes the parameters of the underlying backbone model by minimizing the NT-Xent loss [1]. Similar to [1], we take different samples from the same minibatch as the negative samples. That is, different views of the same sample are considered positive pairs, while views generated from different samples are considered negative pairs.

**MoCoV3** [2] is a SOTA contrastive learning framework for Vision Transformers (ViT). It leverages a query encoder $f_q$ and a key momentum encoder $f_k$ on two stochastically augmented views of a sample to output a query vector $q$ and a key vector $k$. It uses random batch sampling and learns by maximizing the agreement between the positive encoded query and an encoded key pair. In the latest version of MoCo (V3), for a given query $q$, the positive key $k^+$ is encoded from the same sample as $q$, while the negative labels $k^-$ are encoded keys of other samples within the same mini-batch. Both encoders have a similar structure including a backbone plus a projection head, and the query encoder $f_q$ has an additional projection head at the end. The key momentum encoder $f_k$ is slowly updated by a query momentum with the query encoder $f_q$.

**CMC** [20] is a contrastive learning framework focusing on learning from multiview observations. It learns meaningful data representations by contrasting the encoded features from different modalities. To achieve this, it maximizes the agreement between the synchronized representations of different modalities. For each randomly sampled batch with a random augmentation, the backbone model extracts vector representations of each modality. Then, for each pair of modalities, we maximize the similarity between modality representations of the same samples and regard mismatched modality representations from different samples as negative pairs. We sum up the losses for all pairs of modalities to optimize the backbone parameters. For downstream tasks, a linear classification layer is applied on top of concatenated modality representations.

**MAE** [6] is a self-supervised learning approach based on the auto-encoding paradigm. It incorporates the Transformer architecture and achieves SOTA performance on multiple vision tasks. Unlike contrastive learning, MAE does not depend heavily on random augmentations. During the pretraining, we randomly mask a significant portion (*i.e.*, 75%) of each modality input. Instead of dropping the masked patches as in the original MAE paper, we replace them with 0 values to ensure consistent dimensions for the Swin-Transformer and DeepSense operations. A separate encoder and decoder are used for each modality. Before encoding, the modality spectrogram is first projected into fixed-size (*e.g.*, 2x2) patches through a convolutional layer, on top of which the modality embeddings are extracted by the modality encoder. Between independent modality encoding and decoding, we first apply multiple fully-connected layers to the concatenated modality features for modality information fusion and then use separate MLP projection layers to get the projected modality embeddings before decoding. This step is created to enable interactions between modalities. Finally, the modality decoder reconstructs the modality input from the projected modality embeddings. The overall objective is to minimize the mean squared error (MSE) between the original modality patches and the reconstructed modality patches on the masked locations. During the inference, the modality decoders are dropped and only modality encoders are used to extract the latent representations from unmasked modality input. In the end, a linear classification layer is applied to the concatenated modality embeddings to serve the downstream task.

**Cosmo** [14] focuses on contrastive fusion learning from multimodal time-series data to extract modality-consistent information. Cosmo applies separate modality encoders to extract the embedding vector of each modality from the randomly sampled mini-batches. After encoding, each modality embedding is mapped to a hypersphere through an MLP projector and a normalization layer. Then, Cosmo applies a fusion-based feature augmentation to generate $P$ randomly combined features by multiplying the modality embeddings with $P$ normalized random weight vectors. When calculating contrastive loss, these $P$ fusion-based augmented features are considered as positive pairs, while features generated through the same approach but from different samples are treated as negative pairs.

**Cocoa** [3] extends the self-supervised learning of multimodal sensing data by exploring both the cross-modal correlation and intra-modal separation. Similar to other modality-level contrastive frameworks, Cocoa applies a separate backbone encoder to extract the latent embedding of each modality from the randomly sampled and augmented mini-batch. Cocoa has two losses: Cross-modality correlation loss and discriminator loss. Cross-modality correlation loss maximizes the consistency between different modality embeddings corresponding to the same sample by defining them as hard positive pairs. On the contrary, discriminator loss tends to minimize the agreement within a modality, by separating modality embeddings of irrelevant samples within the mini-batch from each other.

**GMC** [15] introduces a multimodal contrastive loss function that encourages the geometric alignment of different modality embeddings. Similar to other multimodal contrastive frameworks, samples are randomly batched and augmented. GMC consists of modality-specific encoders and a joint encoder that simultaneously takes all modality data as input. An additional linear layer is used to map the joint embedding to the same space as individual modality embeddings. Then, a shared projection head is then employed to project both the modality embeddings and the joint embeddings before calculating the contrastive loss. To align the local views (*i.e.*, individual modality embeddings) with the global view (*i.e.*, joint embeddings) in a context-aware manner, GMC minimizes a multimodal contrastive NT-Xent loss by defining the modality-specific embeddings and joint embeddings of the same samples as positive pairs, while treating local-global embedding pairs from different samples as negative pairs.

**MTSS** [17] is a predictive self-supervised learning framework by exploiting the distinguishability among different data transformations. It uses random augmentation ID prediction as the pretext task during the pretraining. Specifically, MTSS first formulates random batches and applies random augmentation to either time or frequency domain. Each modality is augmented with the selected random augmentation with a probability of 50%. Then, individual modality encoders extract modality embeddings from their input, followed by modality fusion to compute the overall sample embeddings. Different from contrastive frameworks, a shallow classifier is included to classify "which random augmentation is applied to the input". A cross-entropy loss is calculated between the predicted augmentation ID and the actual augmentation ID as the pertaining objective. For downstream tasks, only the backbone sample encoder (including the modality encoder and modality fusion layers) is used to extract the sample embeddings, along with a linear classification layer appended at the end of the sample encoder.

**TS2Vec** [24] proposes to learn representations of time series by simultaneously performing temporal contrastive tasks and instance contrastive tasks at multiple granularities (*i.e.*, lengths of sample windows). Instead of creating random batch samples, TS2Vec involves randomly sampled sequences in each batch, with each sequence containing temporally close samples. TS2Vec employs a hierarchical contrasting method to learn representations at multiple sample window granularities. It always regards the same sample under different augmentations and sequence contexts as the positive pairs, while in the instance contrastive task, different samples from separate sequences are regarded as negative pairs, and in the temporal contrastive task, different samples within the same sequence are regarded as negative pairs. At each sample window level, TS2Vec computes both the temporal contrastive loss and instance discrimination loss.

**TNC** [21] learns time series representations with a debiased contrastive objective to distinguish samples within the temporal neighborhood from temporally distant samples. It utilizes a backbone encoder to extract the feature representations from the time series data in a randomly sampled sequence batch. For each sample, TNC identifies a group of samples with similar timestamps as neighboring samples and a group of distant samples as non-neighboring samples. In this paper, we consider samples within the same sequence as the neighboring samples and samples from different sequences as non-neighboring samples. A discriminator is used to learn the time series distribution by predicting the probability of each sample and its neighboring/non-neighboring samples being in the same window. The objective is to maximize the similarity of neighboring samples while pushing the similarity of non-neighboring samples to zero.

**TS-TCC** [4] learns robust representation by performing cross-view predictions and contrasting both temporal and contextual information. It randomly groups multiple sequences into a mini-batch. It first generates two views through random augmentations on each sample. For each view, it extracts context vectors of each timestamp from all sample representations up to this timestamp within the sequence with an autoregressive model and then uses the context vectors from one view to predict

Table 6: DeepSense Configurations.

| Dataset | MOD | ACIDS | RealWorld-HAR | PAMAP2 |
|---|---|---|---|---|
| Dropout Ratio | 0.2 | 0.2 | 0.2 | 0.2 |
| Mod Conv Kernel | aud: [1, 5], sei: [1,3] | [1,4] | [1, 3] | [1, 5] |
| Mod Conv Channel | 128 | 128 | 128 | 64 |
| Mod Conv Layers | 5 | 6 | 6 | 4 |
| Recurrent Dim | 256 | 128 | 256 | 64 |
| Recurrent Layers | 2 | 2 | 2 | 2 |
| FC Dim | 512 | 256 | 256 | 128 |

the future timesteps of the other view. In the temporal contrastive task, given cross-view predicted representations at a future timestamp, it regards the true future representation at that timestamp from the same sequence as the positive pair and regards samples at that timestamp from other sequences as negative pairs. In the contextual contrastive task, TS-TCC calculates NT-Xent loss by considering different augmentations of the same sample as positive pairs and considering different samples within the same mini-batch as negative pairs.

## D    Backbone Models

We tested with two different backbone encoders in this paper: DeepSense and Swin-Transformer (SW-T for short). Both models process the spectrogram of each input sensing modality separately, before the information fusion between the sensing modalities. For each backbone model, the configuration is tuned to achieve the best-supervised model accuracy.

**DeepSense [23]**: It is a state-of-the-art neural network model for time-series sensing data processing. Given the time-frequency spectrogram of each sensing modality, it first uses stacked convolutional layers to extract localized modality features within each time interval. Then, modality information fusion is performed by taking the mean of flattened modality features. Finally, the features across time intervals are aggregated through recurrent layers (*e.g.*, Gated Recurrent Unit (GRU)). For learning frameworks that operate on modality-level features (*i.e.*, FOCAL, CMC, Cosmo, Cocoa, and MAE), we skip the mean fusion among modalities and use individual recurrent layers for each modality, before calculating the pretrain loss.

**Swin-Transformer (SW-T) [11]**: It is a state-of-the-art Transformer model for processing image data. We adapt it to process the time-frequency spectrogram input. Similar to convolution operations, it adaptively allocates attention within subframe windows of input with hierarchical resolutions. The modality input is first partitioned into patches with a convolutional layer. Then, it gradually extracts features from local and shifted windows with multiple blocks. The shift window operation is introduced to break the boundary of partitioned windows and increase the perception area of each window. Each block consists of multiple self-attention layers. The patch resolution of the feature map is halved at the end of each block by merging neighboring patches while the channel number is doubled, such that the receptive field increases as going into deeper layers while the number of patches within each window is fixed. A separate SW-T encoder is used to extract features from each modality input, after which a stack of self-attention layers is appended for information fusion from multiple modalities. Similarly, for learning frameworks that operate on modality-level features, we skip the attention-based fusion blocks and directly calculate pretrain losses on top of modality features.

## E    Training Configurations

In this section, we detail the training strategies used in this paper, which are summarized in Table 8. For each framework, the same configuration is mostly shared between different backbone encoders with few exceptions.

During the pertaining, we use the AdamW [12] optimizer with the cosine schedules [13]. The start learning rate is tuned accordingly for each framework according to their convergence situation. We did observe Cosmo [14] is hard to converge in some cases thus we have to reduce its start learning rate. The used batch size is 256, where 64 short sequences of 4 samples are randomly selected in each batch. The constitution of sequences is determined at the initialization and does not change over

Table 7: Swin-Transformer Configurations.

| Dataset | MOD | ACIDS | RealWorld-HAR | PAMAP2 |
|---|---|---|---|---|
| Dropout Ratio | 0.2 | 0.2 | 0.2 | 0.2 |
| Patch Size | aud: [1, 40], sei: [1,1] | [1, 8] | [1, 2] | [1, 2] |
| Window Size | [3, 3] | [2,4] | [3, 3] | [3, 5] |
| Mod Feature Block Num | [2, 2, 4] | [2, 2, 4] | [2, 2, 2] | [2, 2, 2] |
| Mod Feature Block Channels | [64, 128, 256] | [64, 128, 256] | [32, 64, 128] | [32, 64, 128] |
| Head Num | 4 | 4 | 4 | 4 |
| Mod Fusion Channel | 256 | 256 | 128 | 128 |
| Mod Fusion Head Num | 4 | 4 | 4 | 4 |
| Mod Fusion Block | 2 | 2 | 2 | 2 |
| FC Dim | 512 | 512 | 256 | 128 |

Table 8: Training configurations. (We use LR for Learning Rate)

| Dataset | MOD | ACIDS | RealWorld-HAR | PAMAP2 |
|---|---|---|---|---|
| Temperature | 0.07 | 0.2 | 0.07 | 0.07 |
| Batch Size | 256 | 256 | 256 | 256 |
| Sequence Length | 4 | 4 | 4 | 4 |
| Pretrain Optimizer | AdamW | AdamW | AdamW | AdamW |
| Pretrain Max LR | Default: 1e-4
Cosmo, TNC, GMC, TS2Vec, TSTCC: 1e-5 | Default: 1e-4
Cosmo: 1e-5 | Default: 1e-4
CMC, GMC: 5e-4
Cosmo: 1e-5 | Default: 1e-4
CMC, GMC: 5e-4
Cosmo: 1e-5 |
| Pretrain Min LR | 1e-07 | 1e-07 | 1e-07 | 1e-07 |
| Pretrain Scheduler | Cosine | Cosine | Cosine | Cosine |
| Pretrain Epochs | 6000 | 3000 | 1000 | 1000 |
| Pretrain Weight Decay | 0.05 | 0.05 | 0.05 | 0.05 |
| Finetune Optimizer | Adam | Adam | Adam | Adam |
| Finetune Start LR | 0.001 | 0.0003 | 0..001 | 0.001 |
| Finetune Scheduler | step | step | step | step |
| Finetune LR Decay | 0.2 | 0.2 | 0.2 | 0.2 |
| Finetune LR Period | 50 | 50 | 50 | 50 |
| Finetune Epochs | 200 | 200 | 200 | 200 |

training epochs. The temperature is tuned to achieve the best linear classification performance after the finetuning. A weight decay of 0.05 is used as the training regularization.

During the finetuning, we use the Adam [9] optimizer with the step scheduler. Essentially, the learning rate decays by 0.2 at the end of each period. By default, finetuning runs for 200 epochs in total, and each period is 50 epochs. Besides, the weight decay parameter is separately tuned for each framework for the best balance between training fit and validation fit.

The models are trained on a lab workstation with AMD Threadripper PRO 3000WX Processor of 64 cores and NVIDIA RTX 3090 GPUs. The implementation is based on PyTorch 1.14, and the pretraining on a single GPU spans between 3 hours to 4 days among different datasets and backbone encoders.

# F  Additional Evaluation Results

In this section, we report additional evaluation results and analyses that are not included in the main paper.

## F.1  Finetuning: Complete Linear Classification Results

**Setup:** For each dataset, we apply two backbone encoders (DeepSense and SW-T), and finetune the linear classifier with three different ratios of available labels (100%, 10%, and 1%). For label ratios 10% and 1%, we take 5 random portions of labels for finetuning in each training framework and report the mean and standard deviation among the runs with all testing data. The best result under each configuration is highlighted with the **bold** text. Besides, we also train a supervised model for each configuration as a reference to the self-supervised frameworks.

**Analysis:** Table 9, Table 10, Table 11, and Table 12 summarize the complete linear finetuning results on MOD, ACIDS, RealWorld-HAR, and PAMAP2 datasets, respectively.

Table 9: Fintuning Experiments with Linear Classifier on MOD dataset.

| Encoder | Framework | Label Ratio: 1.0 | | Label Ratio: 0.1 | | Label Ratio: 0.01 | |
|---------|-----------|------|------|------|------|------|------|
| | | Acc | F1 | Acc | F1 | Acc | F1 |
| DeepSense | Supervised | 0.9404 | 0.9399 | 0.6821 ± 0.0442 | 0.6810 ± 0.0475 | 0.3567 ± 0.0450 | 0.3366 ± 0.0365 |
| | SimCLR | 0.8855 | 0.8855 | 0.8186 ± 0.0055 | 0.8162 ± 0.0058 | 0.5934 ± 0.0319 | 0.5808 ± 0.0337 |
| | MoCo | 0.8808 | 0.8812 | 0.7819 ± 0.0078 | 0.7763 ± 0.0089 | 0.5038 ± 0.0377 | 0.4794 ± 0.0509 |
| | CMC | 0.9196 | 0.9186 | 0.8938 ± 0.0055 | 0.8920 ± 0.0056 | 0.7645 ± 0.0131 | 0.7459 ± 0.0224 |
| | MAE | 0.5981 | 0.5993 | 0.4963 ± 0.0083 | 0.4985 ± 0.0041 | 0.3586 ± 0.0347 | 0.3292 ± 0.0497 |
| | Cosmo | 0.8989 | 0.8998 | 0.8505 ± 0.0066 | 0.8519 ± 0.0061 | 0.7025 ± 0.0169 | 0.7025 ± 0.0171 |
| | Cocoa | 0.8774 | 0.8764 | 0.8397 ± 0.0058 | 0.8378 ± 0.0055 | 0.7181 ± 0.0198 | 0.6998 ± 0.0226 |
| | MTSS | 0.4153 | 0.3582 | 0.3863 ± 0.0058 | 0.3139 ± 0.0081 | 0.3140 ± 0.0084 | 0.2527 ± 0.0198 |
| | TS2Vec | 0.7669 | 0.7648 | 0.7018 ± 0.0066 | 0.6980 ± 0.0070 | 0.5319 ± 0.0199 | 0.5150 ± 0.0230 |
| | GMC | 0.9257 | 0.9267 | 0.8812 ± 0.0061 | 0.8820 ± 0.0069 | 0.7198 ± 0.0097 | 0.6983 ± 0.0204 |
| | TNC | 0.9518 | 0.9528 | 0.9437 ± 0.0055 | 0.9446 ± 0.0054 | **0.8616 ± 0.0330** | 0.8469 ± 0.0620 |
| | TSTCC | 0.8707 | 0.8735 | 0.8295 ± 0.0034 | 0.8319 ± 0.0036 | 0.6080 ± 0.0321 | 0.5753 ± 0.0553 |
| | FOCAL | **0.9732** | **0.9729** | **0.9485 ± 0.0038** | **0.9480 ± 0.0039** | 0.8567 ± 0.0151 | **0.8544 ± 0.0173** |
| SW-T | Supervised | 0.8948 | 0.8931 | 0.5555 ± 0.0164 | 0.5450 ± 0.0197 | 0.2028 ± 0.0111 | 0.1638 ± 0.0196 |
| | SimCLR | 0.9250 | 0.9247 | 0.8891 ± 0.0040 | 0.8888 ± 0.0042 | 0.7523 ± 0.0368 | 0.7443 ± 0.0442 |
| | MoCo | 0.9390 | 0.9384 | 0.9073 ± 0.0032 | 0.9073 ± 0.0032 | 0.7482 ± 0.0228 | 0.7409 ± 0.0269 |
| | CMC | 0.9129 | 0.9105 | 0.8691 ± 0.0067 | 0.8661 ± 0.0067 | 0.6994 ± 0.0157 | 0.6835 ± 0.0191 |
| | MAE | 0.7803 | 0.7772 | 0.6561 ± 0.0119 | 0.6480 ± 0.0120 | 0.3764 ± 0.0200 | 0.3544 ± 0.0297 |
| | Cosmo | 0.3429 | 0.3378 | 0.2122 ± 0.0087 | 0.1989 ± 0.0071 | 0.1753 ± 0.0152 | 0.1346 ± 0.0138 |
| | Cocoa | 0.7040 | 0.7038 | 0.6869 ± 0.0145 | 0.6833 ± 0.0177 | 0.6122 ± 0.0162 | 0.5955 ± 0.0300 |
| | MTSS | 0.4206 | 0.4163 | 0.3799 ± 0.0087 | 0.3700 ± 0.0081 | 0.3113 ± 0.0259 | 0.2964 ± 0.0191 |
| | TS2Vec | 0.7254 | 0.7174 | 0.6522 ± 0.0086 | 0.6434 ± 0.0099 | 0.4750 ± 0.0225 | 0.4477 ± 0.0355 |
| | GMC | 0.8640 | 0.8611 | 0.7712 ± 0.0049 | 0.7685 ± 0.0053 | 0.5191 ± 0.0209 | 0.4959 ± 0.0348 |
| | TNC | 0.8533 | 0.8539 | 0.8436 ± 0.0068 | 0.8443 ± 0.0070 | 0.7996 ± 0.0331 | 0.7935 ± 0.0419 |
| | TSTCC | 0.8734 | 0.8735 | 0.8564 ± 0.0040 | 0.8558 ± 0.0038 | 0.7473 ± 0.0220 | 0.7322 ± 0.0470 |
| | FOCAL | **0.9805** | **0.9800** | **0.9593 ± 0.0025** | **0.9584 ± 0.0024** | **0.8840 ± 0.0299** | **0.8776 ± 0.0389** |

First, FOCAL consistently demonstrates significant improvements in both accuracy and F1 score across all label ratios compared to other self-supervised learning baselines on the ACIDS, RealWorld-HAR, and PAMAP2 datasets. In the case of the MOD dataset under 1% labels, FOCAL achieves similar accuracy to TNC with the DeepSense encoder but beats TNC by 10.56% with the SW-T encoder. These results underline the superior performance of FOCAL in multimodal time series sensing data and emphasize the importance of the underlying relationship between the shared and private modality features through time.

Second, the performance improvements persist across backbone encoders and different label ratios, proving the advantage of FOCAL in improving the label efficiency during downstream finetuning. Although there are a few cases where some baselines perform close to FOCAL (*e.g.*, TNC with DeepSense encoder on MOD dataset under 1% labels), such comparability does not persist across encoders.

Third, FOCAL shows comparable performance to the supervised model when all available labels (*i.e.*, 100%) are used in the training. However, when fewer labels are available, FOCAL shows a larger advantage over the supervised oracle, demonstrating its capability to better leverage the limited available labels in adapting to downstream tasks. On average, FOCAL surpasses the supervised model by 1.37% with 100% labels, 15.04% with 10% labels, and 68.39% with 1% labels. By learning semantically meaningful multimodal representations from the massive unlabeled inputs during the pretraining phase, FOCAL can effectively utilize limited data labels during the finetuning process. This is especially reflected in the MOD results, where we have around 6 times more data in pretraining than the finetuning and achieve 3.49% and 9.58% improvement over the supervised model.

Fourth, between the backbone encoders, we found FOCAL brings more relative performance improvement to SW-T than DeepSense compared to their supervised versions. With FOCAL training, SW-T beats DeepSense in two out of four datasets (*i.e.*, MOD and RealWorld-HAR), while DeepSense is always the better encoder architecture with supervised training. Besides, the performance improvement on SW-T is more significant when the number of available labels is low during the finetuning (*i.e.*, 10% and 1%) since larger performance gaps are observed between FOCAL and supervised models.

Table 10: Fintuning Experiments with Linear Classifier on ACIDS dataset.

| Encoder | Framework | Label Ratio: 1.0 | | Label Ratio: 0.1 | | Label Ratio: 0.01 | |
|---|---|---|---|---|---|---|---|
| | | Acc | F1 | Acc | F1 | Acc | F1 |
| | Supervised | **0.9566** | 0.8407 | **0.9379 ± 0.0158** | 0.8006 ± 0.0316 | 0.7567 ± 0.0335 | 0.5754 ± 0.0406 |
| DeepSense | SimCLR | 0.7438 | 0.6101 | 0.7111 ± 0.0157 | 0.5773 ± 0.0166 | 0.6166 ± 0.0206 | 0.4392 ± 0.0430 |
| | MoCo | 0.7717 | 0.6205 | 0.7433 ± 0.0269 | 0.5833 ± 0.0243 | 0.6637 ± 0.0414 | 0.4827 ± 0.0470 |
| | CMC | 0.8443 | 0.7244 | 0.7370 ± 0.0126 | 0.6139 ± 0.0180 | 0.6313 ± 0.0633 | 0.4726 ± 0.0786 |
| | MAE | 0.6644 | 0.5618 | 0.5862 ± 0.0024 | 0.4479 ± 0.0062 | 0.4901 ± 0.0309 | 0.2825 ± 0.0293 |
| | Cosmo | 0.8511 | 0.6929 | 0.8532 ± 0.0176 | 0.7083 ± 0.0199 | 0.7288 ± 0.0231 | 0.5571 ± 0.0447 |
| | Cocoa | 0.6644 | 0.5359 | 0.6174 ± 0.0106 | 0.4605 ± 0.0219 | 0.5617 ± 0.0223 | 0.3811 ± 0.0289 |
| | MTSS | 0.4352 | 0.2441 | 0.4247 ± 0.0341 | 0.2130 ± 0.0385 | 0.4280 ± 0.0274 | 0.1879 ± 0.0333 |
| | TS2Vec | 0.5224 | 0.3587 | 0.5299 ± 0.0121 | 0.3554 ± 0.0113 | 0.5341 ± 0.0363 | 0.3516 ± 0.0366 |
| | GMC | 0.9096 | 0.7929 | 0.8890 ± 0.0090 | 0.7681 ± 0.0178 | 0.7156 ± 0.0603 | 0.5573 ± 0.0693 |
| | TNC | 0.8237 | 0.6936 | 0.8063 ± 0.0156 | 0.6635 ± 0.0370 | 0.7428 ± 0.0419 | 0.5760 ± 0.0576 |
| | TSTCC | 0.7667 | 0.6164 | 0.7655 ± 0.0094 | 0.6127 ± 0.0083 | 0.6697 ± 0.0354 | 0.4846 ± 0.0368 |
| | FOCAL | 0.9516 | **0.8580** | 0.9253 ± 0.0143 | **0.8007 ± 0.0199** | **0.7829 ± 0.0448** | **0.5940 ± 0.0514** |
| | Supervised | 0.9137 | 0.7770 | 0.7310 ± 0.0224 | 0.5532 ± 0.0158 | 0.2666 ± 0.0319 | 0.1531 ± 0.0398 |
| SW-T | SimCLR | 0.9128 | 0.8144 | 0.8882 ± 0.0154 | 0.7751 ± 0.0161 | 0.7580 ± 0.0380 | 0.6030 ± 0.0565 |
| | MoCo | 0.9174 | 0.8100 | 0.9069 ± 0.0111 | 0.7841 ± 0.0192 | 0.7990 ± 0.0299 | 0.6235 ± 0.0408 |
| | CMC | 0.8128 | 0.6857 | 0.7985 ± 0.0129 | 0.6700 ± 0.0170 | 0.6583 ± 0.0401 | 0.4990 ± 0.0422 |
| | MAE | 0.8516 | 0.7023 | 0.7916 ± 0.0066 | 0.6344 ± 0.0088 | 0.4751 ± 0.0631 | 0.3440 ± 0.0317 |
| | Cosmo | 0.7110 | 0.6086 | 0.6722 ± 0.0102 | 0.5279 ± 0.0067 | 0.5419 ± 0.0235 | 0.3710 ± 0.0114 |
| | Cocoa | 0.7096 | 0.5794 | 0.6711 ± 0.0117 | 0.5324 ± 0.0127 | 0.6262 ± 0.0282 | 0.4585 ± 0.0212 |
| | MTSS | 0.3429 | 0.2250 | 0.2878 ± 0.0292 | 0.1782 ± 0.0113 | 0.2946 ± 0.0499 | 0.1564 ± 0.0142 |
| | TS2Vec | 0.7183 | 0.5748 | 0.6756 ± 0.0124 | 0.5003 ± 0.0119 | 0.5801 ± 0.0194 | 0.3837 ± 0.0153 |
| | GMC | 0.9402 | 0.7766 | 0.9014 ± 0.0116 | 0.7278 ± 0.0148 | 0.7089 ± 0.0426 | 0.5250 ± 0.0401 |
| | TNC | 0.8352 | 0.7372 | 0.8158 ± 0.0135 | 0.7051 ± 0.0176 | 0.6827 ± 0.0469 | 0.5424 ± 0.0500 |
| | TSTCC | 0.9041 | 0.7547 | 0.9009 ± 0.0062 | 0.7449 ± 0.0202 | 0.7656 ± 0.0378 | 0.5806 ± 0.0223 |
| | FOCAL | **0.9489** | **0.8262** | **0.9400 ± 0.0081** | **0.7975 ± 0.0199** | **0.8669 ± 0.0287** | **0.6844 ± 0.0372** |

Table 11: Fintuning Experiments with Linear Classifier on RealWorld-HAR dataset.

| Encoder | Framework | Label Ratio: 1.0 | | Label Ratio: 0.1 | | Label Ratio: 0.01 | |
|---|---|---|---|---|---|---|---|
| | | Acc | F1 | Acc | F1 | Acc | F1 |
| | Supervised | 0.9348 | **0.9388** | 0.9256 ± 0.0056 | **0.9233 ± 0.0104** | 0.7305 ± 0.0270 | 0.6158 ± 0.0341 |
| DeepSense | SimCLR | 0.7138 | 0.6841 | 0.6597 ± 0.0182 | 0.6126 ± 0.0198 | 0.5334 ± 0.0566 | 0.4271 ± 0.0518 |
| | MoCo | 0.7859 | 0.7708 | 0.7454 ± 0.0206 | 0.6687 ± 0.0340 | 0.5110 ± 0.0409 | 0.4018 ± 0.0552 |
| | CMC | 0.7975 | 0.8116 | 0.7482 ± 0.0328 | 0.7590 ± 0.0282 | 0.5169 ± 0.0314 | 0.4716 ± 0.0455 |
| | MAE | 0.7565 | 0.7515 | 0.7206 ± 0.0181 | 0.7056 ± 0.0175 | 0.5556 ± 0.0527 | 0.4593 ± 0.0541 |
| | Cosmo | 0.8956 | 0.8888 | 0.8814 ± 0.0123 | 0.8626 ± 0.0338 | 0.8434 ± 0.0376 | 0.7775 ± 0.0801 |
| | Cocoa | 0.8465 | 0.8488 | 0.8492 ± 0.0070 | 0.8211 ± 0.0068 | 0.7155 ± 0.0397 | 0.6381 ± 0.0324 |
| | MTSS | 0.2989 | 0.1405 | 0.1905 ± 0.0503 | 0.0692 ± 0.0328 | 0.1698 ± 0.0365 | 0.0600 ± 0.0355 |
| | TS2Vec | 0.6595 | 0.5984 | 0.6419 ± 0.0189 | 0.5721 ± 0.0154 | 0.6147 ± 0.0456 | 0.5197 ± 0.0241 |
| | GMC | 0.8869 | 0.8948 | 0.8872 ± 0.0172 | 0.8842 ± 0.0124 | 0.7954 ± 0.0367 | 0.7620 ± 0.0442 |
| | TNC | 0.8892 | 0.8971 | 0.8712 ± 0.0238 | 0.8629 ± 0.0260 | 0.7991 ± 0.0390 | 0.7337 ± 0.0229 |
| | TSTCC | 0.8073 | 0.8010 | 0.7892 ± 0.0146 | 0.7625 ± 0.0223 | 0.7213 ± 0.0320 | 0.6181 ± 0.0352 |
| | FOCAL | **0.9382** | 0.9290 | **0.9335 ± 0.0053** | 0.9224 ± 0.0075 | **0.8518 ± 0.0274** | **0.7933 ± 0.0436** |
| | Supervised | 0.9313 | 0.9278 | 0.7264 ± 0.0411 | 0.6090 ± 0.0447 | 0.4541 ± 0.0694 | 0.2771 ± 0.0798 |
| SW-T | SimCLR | 0.7046 | 0.7220 | 0.6717 ± 0.0062 | 0.6892 ± 0.0081 | 0.4867 ± 0.0431 | 0.4267 ± 0.0674 |
| | MoCo | 0.7813 | 0.8024 | 0.7324 ± 0.0096 | 0.7425 ± 0.0173 | 0.5541 ± 0.0462 | 0.4823 ± 0.0391 |
| | CMC | 0.8840 | 0.8955 | 0.8352 ± 0.0154 | 0.8424 ± 0.0156 | 0.5602 ± 0.0411 | 0.5245 ± 0.0549 |
| | MAE | 0.8829 | 0.8813 | 0.7873 ± 0.0100 | 0.7224 ± 0.0314 | 0.5602 ± 0.0275 | 0.4699 ± 0.0205 |
| | Cosmo | 0.8604 | 0.8169 | 0.7710 ± 0.0134 | 0.6899 ± 0.0178 | 0.6089 ± 0.0256 | 0.5230 ± 0.0395 |
| | Cocoa | 0.8892 | 0.8861 | 0.8609 ± 0.0110 | 0.8501 ± 0.0143 | 0.7430 ± 0.0321 | 0.6657 ± 0.0432 |
| | MTSS | 0.5136 | 0.4370 | 0.4359 ± 0.0281 | 0.3690 ± 0.0303 | 0.3547 ± 0.0156 | 0.2792 ± 0.0202 |
| | TS2Vec | 0.6151 | 0.5955 | 0.6074 ± 0.0202 | 0.5540 ± 0.0201 | 0.5667 ± 0.0451 | 0.4876 ± 0.0464 |
| | GMC | 0.9319 | 0.9379 | 0.9081 ± 0.0108 | 0.9115 ± 0.0092 | 0.7925 ± 0.0426 | 0.7453 ± 0.0581 |
| | TNC | 0.8817 | 0.8784 | 0.8635 ± 0.0109 | 0.8525 ± 0.0100 | 0.8061 ± 0.0215 | 0.7494 ± 0.0452 |
| | TSTCC | 0.8731 | 0.8454 | 0.8606 ± 0.0114 | 0.8070 ± 0.0233 | 0.7374 ± 0.0434 | 0.6685 ± 0.0642 |
| | FOCAL | **0.9452** | **0.9492** | **0.9370 ± 0.0069** | **0.9421 ± 0.0060** | **0.8301 ± 0.0428** | **0.7519 ± 0.0578** |

## F.2 Finetuning: Complete KNN Classification Results

**Setup:** In addition to linear probing, we further evaluate the self-supervised frameworks on four datasets using the K-Nearest-Neighbors (KNN, K=5) classifier without introducing new parameters. This evaluation method allows us to examine the quality of learned representations without new

Table 12: Fintuning Experiments with Linear Classifier on PAMAP2 dataset.

| Encoder | Framework | Label Ratio: 1.0 | | Label Ratio: 0.1 | | Label Ratio: 0.01 | |
|---|---|---|---|---|---|---|---|
| | | Acc | F1 | Acc | F1 | Acc | F1 |
| DeepSense | Supervised | **0.8849** | **0.8761** | 0.8080 ± 0.0071 | 0.7649 ± 0.0275 | 0.6539 ± 0.0303 | 0.5695 ± 0.0726 |
| | SimCLR | 0.6802 | 0.6583 | 0.6132 ± 0.0174 | 0.5606 ± 0.0247 | 0.4352 ± 0.0340 | 0.3305 ± 0.0197 |
| | MoCo | 0.7559 | 0.7387 | 0.6325 ± 0.0177 | 0.5601 ± 0.0401 | 0.3872 ± 0.0301 | 0.2873 ± 0.0274 |
| | CMC | 0.7906 | 0.7706 | 0.6687 ± 0.0263 | 0.5653 ± 0.0602 | 0.2724 ± 0.0287 | 0.1676 ± 0.0248 |
| | MAE | 0.7114 | 0.6158 | 0.5769 ± 0.0222 | 0.4514 ± 0.0239 | 0.2734 ± 0.0192 | 0.1096 ± 0.0198 |
| | Cosmo | 0.8356 | 0.8135 | 0.7790 ± 0.0220 | 0.7427 ± 0.0341 | 0.6782 ± 0.0226 | 0.5740 ± 0.0293 |
| | Cocoa | 0.7603 | 0.7187 | 0.7132 ± 0.0105 | 0.6432 ± 0.0082 | 0.5922 ± 0.0234 | 0.5293 ± 0.0232 |
| | MTSS | 0.3541 | 0.1795 | 0.2891 ± 0.0416 | 0.1169 ± 0.0378 | 0.1857 ± 0.0546 | 0.0710 ± 0.0406 |
| | TS2Vec | 0.5729 | 0.4715 | 0.5416 ± 0.0171 | 0.4433 ± 0.0177 | 0.4399 ± 0.0341 | 0.3335 ± 0.0445 |
| | GMC | 0.8119 | 0.7860 | 0.7528 ± 0.0097 | 0.6975 ± 0.0207 | 0.5837 ± 0.0367 | 0.4899 ± 0.0510 |
| | TNC | 0.8387 | 0.8143 | 0.8287 ± 0.0022 | 0.8068 ± 0.0059 | 0.7365 ± 0.0414 | 0.6469 ± 0.0682 |
| | TSTCC | 0.7776 | 0.7250 | 0.7489 ± 0.0105 | 0.6401 ± 0.0201 | 0.5348 ± 0.0782 | 0.4368 ± 0.0852 |
| | FOCAL | 0.8604 | 0.8463 | **0.8373 ± 0.0041** | **0.8175 ± 0.0074** | **0.7521 ± 0.0151** | **0.6900 ± 0.0325** |
| SW-T | Supervised | **0.8612** | **0.8384** | 0.7295 ± 0.0135 | 0.6434 ± 0.0230 | 0.4048 ± 0.0337 | 0.3159 ± 0.0271 |
| | SimCLR | 0.7705 | 0.7424 | 0.7307 ± 0.0060 | 0.6871 ± 0.0103 | 0.5416 ± 0.0441 | 0.4708 ± 0.0627 |
| | MoCo | 0.7717 | 0.7313 | 0.7112 ± 0.0203 | 0.6356 ± 0.0331 | 0.4774 ± 0.0220 | 0.3740 ± 0.0301 |
| | CMC | 0.8080 | 0.7901 | 0.6864 ± 0.0259 | 0.4590 ± 0.0131 | 0.1852 ± 0.0221 | 0.1283 ± 0.0127 |
| | MAE | 0.7910 | 0.7606 | 0.6655 ± 0.0067 | 0.6028 ± 0.0129 | 0.3603 ± 0.0416 | 0.2866 ± 0.0402 |
| | Cosmo | 0.7741 | 0.7366 | 0.6702 ± 0.0051 | 0.5958 ± 0.0107 | 0.4555 ± 0.0381 | 0.3870 ± 0.0297 |
| | Cocoa | 0.7689 | 0.7317 | 0.7461 ± 0.0047 | 0.7048 ± 0.0115 | 0.6594 ± 0.0228 | 0.5973 ± 0.0243 |
| | MTSS | 0.2847 | 0.1714 | 0.2558 ± 0.0109 | 0.1585 ± 0.0097 | 0.2133 ± 0.0164 | 0.1265 ± 0.0215 |
| | TS2Vec | 0.6195 | 0.5426 | 0.6001 ± 0.0133 | 0.5249 ± 0.0154 | 0.5051 ± 0.0402 | 0.4123 ± 0.0374 |
| | GMC | 0.8312 | 0.8083 | 0.7686 ± 0.0118 | 0.7297 ± 0.0140 | 0.5704 ± 0.0409 | 0.4965 ± 0.0426 |
| | TNC | 0.8013 | 0.7506 | 0.7921 ± 0.0083 | 0.7380 ± 0.0144 | 0.7222 ± 0.0305 | 0.6378 ± 0.0488 |
| | TSTCC | 0.7997 | 0.7260 | 0.7800 ± 0.0094 | 0.6890 ± 0.0148 | 0.6438 ± 0.0569 | 0.5566 ± 0.0509 |
| | FOCAL | 0.8442 | 0.8287 | **0.8179 ± 0.0117** | **0.7856 ± 0.0177** | **0.7371 ± 0.0332** | **0.6630 ± 0.0410** |

Table 13: Complete KNN Results

| Encoders | Framework | MOD | | ACIDS | | RealWorld-HAR | | PAMAP2 | |
|---|---|---|---|---|---|---|---|---|---|
| | | Acc | F1 | Acc | F1 | Acc | F1 | Acc | F1 |
| DeepSense | SimCLR | 0.8238 | 0.8240 | 0.7402 | 0.5637 | 0.6584 | 0.6234 | 0.6451 | 0.6114 |
| | MoCo | 0.8446 | 0.8444 | 0.7735 | 0.5957 | 0.7496 | 0.7134 | 0.6924 | 0.6766 |
| | CMC | 0.9002 | 0.8989 | 0.7584 | 0.6516 | 0.5216 | 0.5868 | 0.8032 | 0.7938 |
| | MAE | 0.6470 | 0.6451 | 0.7457 | 0.5610 | **0.8794** | **0.8817** | 0.6857 | 0.6427 |
| | Cosmo | 0.8379 | 0.8387 | 0.7986 | 0.6284 | 0.8102 | 0.7817 | 0.8005 | 0.7743 |
| | Cocoa | 0.7910 | 0.7877 | 0.6758 | 0.4966 | 0.7778 | 0.7459 | 0.7129 | 0.6974 |
| | MTSS | 0.3443 | 0.3249 | 0.4333 | 0.2417 | 0.5101 | 0.4384 | 0.3931 | 0.3379 |
| | TS2Vec | 0.6966 | 0.6875 | 0.5726 | 0.3602 | 0.6480 | 0.5832 | 0.5639 | 0.5180 |
| | GMC | 0.8533 | 0.8526 | 0.7411 | 0.6210 | 0.7415 | 0.7560 | 0.7843 | 0.7543 |
| | TNC | 0.9498 | 0.9508 | 0.7813 | 0.6203 | 0.7882 | 0.7565 | 0.7993 | 0.7653 |
| | TSTCC | 0.8607 | 0.8615 | 0.8192 | 0.6443 | 0.7686 | 0.7658 | 0.8032 | 0.7896 |
| | FOCAL | **0.9551** | **0.9544** | **0.9247** | **0.7938** | 0.8205 | 0.8254 | **0.8482** | **0.8378** |
| SW-T | SimCLR | 0.9022 | 0.9021 | 0.8553 | 0.7086 | 0.6532 | 0.6767 | 0.7441 | 0.7178 |
| | MoCo | 0.9344 | 0.9343 | 0.8311 | 0.6943 | 0.7103 | 0.7303 | 0.7082 | 0.6678 |
| | CMC | 0.8305 | 0.8261 | 0.7187 | 0.6355 | 0.5701 | 0.6007 | 0.7709 | 0.7694 |
| | MAE | 0.3389 | 0.3104 | 0.5945 | 0.4194 | 0.6428 | 0.6080 | 0.5517 | 0.4969 |
| | Cosmo | 0.2786 | 0.2621 | 0.5790 | 0.4573 | 0.7086 | 0.6389 | 0.6672 | 0.5874 |
| | Cocoa | 0.5941 | 0.5793 | 0.5311 | 0.4261 | 0.7421 | 0.7496 | 0.7188 | 0.7070 |
| | MTSS | 0.3423 | 0.3376 | 0.3151 | 0.1890 | 0.4882 | 0.4431 | 0.2007 | 0.1649 |
| | TS2Vec | 0.5847 | 0.5718 | 0.6050 | 0.4144 | 0.5580 | 0.5335 | 0.5623 | 0.5040 |
| | GMC | 0.5318 | 0.5180 | 0.7589 | 0.6150 | 0.7380 | 0.7455 | 0.7567 | 0.7401 |
| | TNC | 0.8265 | 0.8263 | 0.7795 | 0.6725 | 0.8009 | 0.7817 | 0.7674 | 0.7189 |
| | TSTCC | 0.8607 | 0.8613 | 0.8356 | 0.6700 | 0.7582 | 0.7512 | 0.7780 | 0.7369 |
| | FOCAL | **0.9665** | **0.9664** | **0.8826** | **0.7643** | **0.8586** | **0.8665** | **0.8549** | **0.8484** |

training steps. We first construct a KNN estimator using the encoded sample features and corresponding labels from finetuning data. For multi-modal frameworks, we directly concatenate modality embeddings as the sample-level representations. Subsequently, the estimator predicts the test labels according to the labels of neighboring samples in the supervised set $\mathcal{X}^s$ and computes the testing accuracy accordingly.

**Analysis:** The complete evaluation results with the KNN classifier are reported in Table 13. FOCAL consistently surpasses the performance of other self-supervised learning baselines in most cases. The KNN evaluation results are mostly consistent with the linear classification results, but there are

Table 14: Clustering Evaluation

| Dataset | | MOD | | ACIDS | | RealWorld-HAR | | PAMAP2 | |
|---|---|---|---|---|---|---|---|---|---|
| Encoder | Framework | ARI | NMI | ARI | NMI | ARI | NMI | ARI | NMI |
| DeepSense | CMC | **0.3936 ± 0.0125** | **0.5224 ± 0.0206** | 0.2926 ± 0.0156 | 0.5833 ± 0.0051 | 0.2187 ± 0.1094 | 0.4354 ± 0.1713 | 0.3024 ± 0.0118 | 0.5063 ± 0.0120 |
| | Cosmo | 0.1384 ± 0.0540 | 0.2552 ± 0.0803 | 0.5217 ± 0.0074 | 0.6416 ± 0.0184 | 0.4231 ± 0.2726 | 0.5318 ± 0.2564 | 0.3583 ± 0.0781 | 0.5212 ± 0.0671 |
| | Cocoa | 0.3502 ± 0.0184 | 0.4444 ± 0.0135 | 0.5453 ± 0.0229 | 0.6767 ± 0.0184 | 0.3385 ± 0.1826 | 0.4792 ± 0.1940 | 0.3493 ± 0.0230 | 0.5091 ± 0.0184 |
| | GMC | 0.1982 ± 0.0674 | 0.3925 ± 0.0416 | 0.2490 ± 0.0403 | 0.5296 ± 0.0150 | 0.3433 ± 0.1836 | 0.4794 ± 0.1978 | 0.3078 ± 0.0194 | 0.5092 ± 0.0221 |
| | **FOCAL** | 0.3929 ± 0.0222 | 0.5067 ± 0.0226 | **0.5723 ± 0.0440** | **0.7213 ± 0.0432** | **0.4400 ± 0.2465** | **0.5545 ± 0.2437** | **0.4759 ± 0.0695** | **0.6037 ± 0.0558** |
| SW-T | CMC | 0.4314 ± 0.2716 | 0.5413 ± 0.2612 | 0.3604 ± 0.0119 | 0.5881 ± 0.0009 | 0.4014 ± 0.0528 | 0.5275 ± 0.0532 | 0.3718 ± 0.0480 | 0.5562 ± 0.0401 |
| | Cosmo | 0.2865 ± 0.1521 | 0.4140 ± 0.1946 | 0.4436 ± 0.0145 | 0.5469 ± 0.0015 | 0.0029 ± 0.0020 | 0.0107 ± 0.0025 | 0.2425 ± 0.0301 | 0.3604 ± 0.0347 |
| | Cocoa | 0.4281 ± 0.2314 | 0.5308 ± 0.2405 | 0.4363 ± 0.0020 | 0.6824 ± 0.0261 | 0.2487 ± 0.0053 | 0.3897 ± 0.0024 | 0.3658 ± 0.0540 | 0.5330 ± 0.0472 |
| | GMC | 0.3973 ± 0.2177 | 0.4940 ± 0.2184 | 0.2055 ± 0.0029 | 0.4971 ± 0.0066 | 0.3050 ± 0.0076 | 0.4342 ± 0.0052 | 0.2794 ± 0.0206 | 0.5044 ± 0.0329 |
| | **FOCAL** | **0.4660 ± 0.2737** | **0.5693 ± 0.2579** | **0.6050 ± 0.1027** | **0.7389 ± 0.0774** | **0.4319 ± 0.0851** | **0.5462 ± 0.0717** | **0.4785 ± 0.0914** | **0.6130 ± 0.0730** |

also a few exceptions. With the SW-T encoder, FOCAL exceeds the best baseline by an average of 4.85%. When using DeepSense as the encoder, FOCAL outperforms the most competitive contrastive framework baseline by 1.18% across all datasets. In the RealWorld-HAR dataset, DeepSense with MAE achieves higher accuracy than FOCAL, but it fails in the linear classification scenario and fails to generalize to other datasets and backbone encoders. In comparison to other contrastive learning baselines, FOCAL still demonstrates its superiority in KNN classification. Between the two encoders on FOCAL, SW-T outperforms DeepSense in three out of four datasets, which further shows the benefits FOCAL brings to SW-T training.

## F.3 Complete Clustering Results

**Setup:** We further evaluate the clustering performance of FOCAL with other multimodal self-supervised learning baselines, including CMC, Cosmo, Cocoa, and GMC. We apply K-means clustering to the encoded embeddings from each framework, by setting the number of clusters equal to the number of unique classes in the testing dataset. As mentioned before, the preferred cluster structure by the SSL frameworks should align well with the underlying ground-truth labels in addition to presenting clear separation among the clusters. Following this objective, we quantitatively assess the clustering performance by independently calculating the Adjusted Rand Index (ARI) and the Normalized Mutual Information (NMI) of each modality to provide an accurate comparison of the alignment between the pretrained clusters and ground-truth classes. ARI evaluates the similarity between the clustering assignments generated by the K-means clusters and the label distribution of the test data. With a value range of -1 to 1, ARI indicates a high degree of agreement between the two clusterings when close to 1, random agreement when close to zero, and a clustering performance worse than random when approaching -1. NMI serves as an external metric for measuring the clustering quality. A score close to 1 indicates a perfect correlation between the clusterings, and a score of 0 demonstrates no mutual information between the clusters. Lastly, we performed t-SNE to qualitatively visualize the sample embeddings after concatenating the modality embeddings.

**Analysis:** In Table 14, we present the clustering results with the average and standard deviation of ARI and NMI across all modalities. As the results show, FOCAL consistently achieves the highest or similar ARI scores in comparison to other multimodal contrastive frameworks. When using SW-T as the encoder, FOCAL outperforms the strongest baseline by an average ARI margin of 8.33% and an average NMI margin of 4%. With DeepSense as the encoder, FOCAL surpasses the best baseline by an average ARI margin of 4.61% and an average NMI margin of 3.35%. Although CMC exhibits comparable performance for the MOD dataset when using DeepSense as an encoder, FOCAL with DeepSense exceeds CMC by an average of 16.8% and 8.47% in ARI and NMI across the four datasets. These results confirm our claim that FOCAL produces higher quality modality representations compared to the baseline multi-modal contrastive frameworks. We also found the general ARI and NMI values are relatively low because there could be multiple perspectives affecting the cluster structures that lead to complicated underlying semantics while we only evaluate one perspective among them.

Figures 10 and 11 represent the t-SNE visualizations of the encoded sample embeddings by FOCAL. We can observe a clear separation between individual clusters on MOD, ACIDS, and RealWorld-HAR, indicating that FOCAL effectively captures the distinct characteristics of each class. However, for the PAMAP2 dataset, we notice various overlaps between different embeddings. This observation suggests that the underlying structure of the PAMAP2 dataset is more challenging to differentiate compared to other datasets, potentially due to similarities among a large number of classes with 18 different physical activities. This discovery is also consistent with our linear probing results,

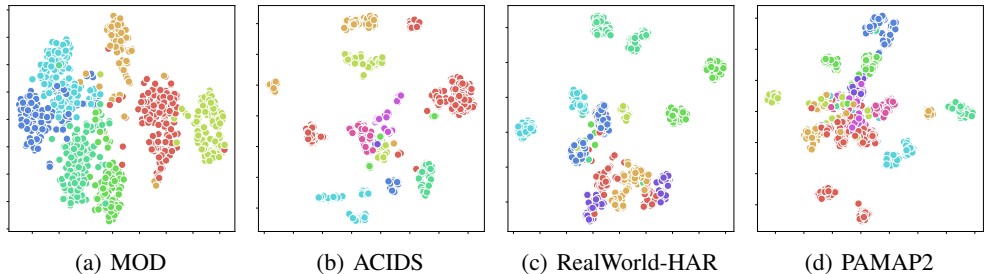

(a) MOD        (b) ACIDS        (c) RealWorld-HAR        (d) PAMAP2

Figure 10: t-SNE visualization of the concatenated modality features in FOCAL with **SW-T encoder**.

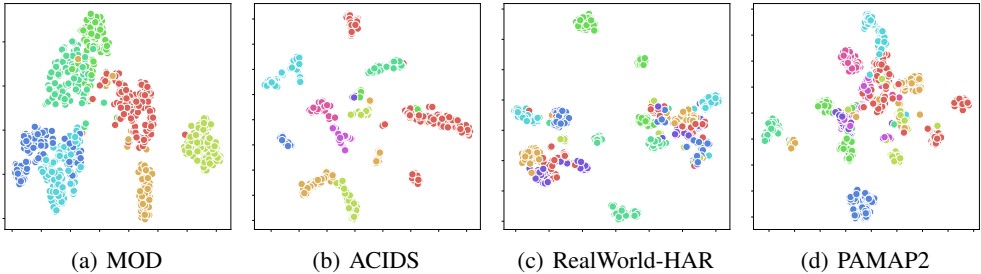

(a) MOD        (b) ACIDS        (c) RealWorld-HAR        (d) PAMAP2

Figure 11: t-SNE visualization of the concatenated modality features in FOCAL with **DeepSense encoder**.

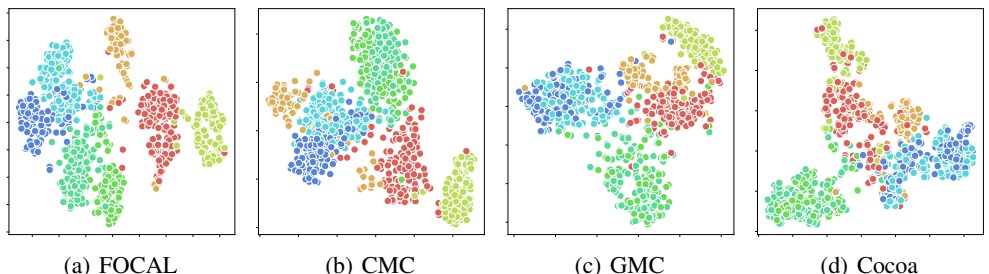

(a) FOCAL        (b) CMC        (c) GMC        (d) Cocoa

Figure 12: t-SNE visualization of the concatenated modality features with SW-T encoder (MOD dataset).

which perform slightly worse on the PAMAP2 dataset. Besides, we provide the t-SNE visualization comparison between FOCAL, CMC, GMC, and Cocoa with Swin-Transformer encoder in Figure 12.

### F.4 Complete Additional Downstream Task Results

**Setup:** We collected additional data samples for the MOD dataset and finetuned our pretrained models from previous experiments. Specifically, we evaluated our pretrained models by finetuning the classifier layer on two downstream tasks, distance classification, and speed classification tasks, with data obtained from different environments and new types of vehicles. These alterations in the data lead to domain adaptation, referring to changes in the data's distribution. For speed classification, the classifier predicts the speed of the moving object between 5, 10, 15, and 20 mph. For distance classification, the classifier outputs whether the detected object is close, near, or far away.

Three metrics are evaluated in this experiment. In addition to the normal accuracy and (macro) F1 score, we also define a new metric called *correlated accuracy*. It considers the semantical distances between different classes and assigns different penalties to different misclassifications cases. Intuitively, for a sample with ground truth speed 5, a misclassification of speed 20 should be assigned more penalty than a misclassification of speed 10. Given a sample label pair $(\mathbf{x}_i^s, y_i^s)$, the predicted label $y_i$, and the number of classes $C$, we define the maximum class distance as $\max(i, \; C - i - 1)$,

Table 15: Linear Finetune Results with Extended Tasks on MOD

| Task | Distance Classification | | | | | | Speed Classification | | | | | |
|---|---|---|---|---|---|---|---|---|---|---|---|---|
| Encoder | SW-T | | | DeepSense | | | SW-T | | | DeepSense | | |
| Framework | Acc | F1 | Corr Acc | Acc | F1 | Corr Acc | Acc | F1 | Corr Acc | Acc | F1 | Corr Acc |
| SimCLR | 0.9090 | 0.8694 | 0.9545 | 0.8787 | 0.8057 | 0.9242 | 0.5511 | 0.5514 | 0.7524 | 0.5596 | 0.5438 | 0.7751 |
| MoCo | 0.9090 | 0.8694 | 0.9545 | 0.8484 | 0.7374 | 0.9091 | 0.6108 | 0.6105 | 0.7879 | 0.5767 | 0.5655 | 0.7794 |
| CMC | 0.8180 | 0.7507 | 0.8636 | **0.9393** | 0.8181 | **0.9697** | 0.5170 | 0.5175 | 0.7268 | 0.6022 | 0.6016 | 0.7850 |
| MAE | 0.7272 | 0.4917 | 0.8333 | 0.7272 | 0.4969 | 0.8030 | 0.4545 | 0.4383 | 0.6932 | 0.4034 | 0.3929 | 0.6506 |
| Cosmo | 0.6363 | 0.2592 | 0.8182 | **0.9393** | 0.8730 | 0.9545 | 0.2926 | 0.2779 | 0.5459 | 0.5681 | 0.5566 | 0.7737 |
| Cocoa | 0.8181 | 0.6898 | 0.8939 | 0.8181 | 0.6966 | 0.8333 | 0.4005 | 0.3618 | 0.6851 | 0.5625 | 0.5580 | 0.7628 |
| MTSS | 0.7272 | 0.4832 | 0.8030 | 0.8787 | 0.6180 | 0.9394 | 0.3522 | 0.2711 | 0.6544 | 0.4005 | 0.3482 | 0.6856 |
| TS2Vec | 0.6969 | 0.5869 | 0.7879 | 0.9090 | 0.8469 | 0.9242 | 0.4517 | 0.4473 | 0.6799 | 0.5198 | 0.5073 | 0.7476 |
| GMC | 0.8181 | 0.7450 | 0.8788 | 0.8484 | 0.7956 | 0.8788 | 0.4460 | 0.4405 | 0.6856 | 0.6250 | 0.6232 | 0.7917 |
| TNC | 0.8484 | 0.8015 | 0.8788 | 0.8787 | 0.8169 | 0.9242 | 0.4375 | 0.4322 | 0.6643 | 0.6108 | 0.6077 | 0.7841 |
| TS-TCC | 0.7878 | 0.6575 | 0.8939 | 0.8484 | 0.7312 | 0.9242 | 0.5284 | 0.5230 | 0.7311 | 0.5255 | 0.5138 | 0.7486 |
| FOCAL | **0.9697** | **0.9726** | **0.9848** | **0.9393** | **0.8985** | **0.9697** | **0.6960** | **0.6920** | **0.8329** | **0.6647** | **0.6682** | **0.8234** |

then the correlated accuracy is calculated by

$$corr\_acc = \frac{1}{N'} \sum_i \left( 1 - \frac{|y_i - y_i^s|}{\max(i, C - i - 1)} \right), \tag{6}$$

where the penalty of misclassification is linearly interpolated according to the distance of the predicted label and the ground truth label, divided by the maximum distance to this class. The value range of the correlated accuracy is still $[0, 1]$, where 0 means the worst and 1 means the best.

**Analysis:** We observe a significant drop in performance on most of the self-supervised learning frameworks on speed classification. When using SW-T as the encoder, FOCAL still dominates the performance over other baselines, exceeding the strongest baseline by 6.07% accuracy and 10.32 % F1 score. When using DeepSense as the encoder, FOCAL also achieves comparable high performance as the current baselines. The advantage of FOCAL persists in the correlated accuracy metric where the physical correlations among classes are counted. Considering the heterogeneous finetune tasks, the potential domain shift, and the leading performance, we conclude that FOCAL is promising in learning fundamental feature patterns from multi-modal sensing data that could serve an extensive set of downstream tasks.

## F.5 Ablation Study Results

**Steup**: We first briefly introduce the compared variants of FOCAL in our ablation study. In these variants, they are set up in the same way as FOCAL except for the places we explain below.

- **FOCAL-noPrivate:** We remove the private modality space and its related contrastive task but only apply the cross-modal matching task.

- **FOCAL-noOrth:** We keep the private modality space, but do not enforce the orthogonality constraint between the shared feature and private feature of the same modality, and the private features between pairs of modalities.

- **FOCAL-wDistInd:** We replace the geometrical orthogonality constraint with statistical independence between modality embedding distributions. Specifically, we follow the approach proposed in [8] to disentangle the distribution of latent subspaces, which minimizes the mutual information between shared-private spaces of the same modality and private-private spaces between two modalities. Given two embedding distributions, it minimizes the KL divergence between their joint distribution and the product of two marginal distributions. Following the density-r atio trick, we train a classifier consisting of several fully-connected layers to discriminate samples from the originally matched pairs of embeddings and the randomly selected embedding pairs, which has been shown to approximate the density ratio needed to estimate the KL divergence within sample batches. Similar to GAN [5], we train the discriminator alternatively with modality encoders until convergence.

- **FOCAL-noTemp:** We remove the temporal structural constraint proposed in FOCAL.

- **FOCAL-wTempCon:** We replace the temporal structural constraint with a temporal contrastive task. Given a modality, we regard close sample pairs within a short sequence as positive samples and regard distant sample pairs from different short sequences as negative samples, and conduct discrimination between positive samples and negative samples.

Table 16: Ablation Results with DeepSense Encoder and Linear Classifier

| Metrics | MOD | | ACIDS | | RealWorld-HAR | | PAMAP2 | |
|---|---|---|---|---|---|---|---|---|
| | Acc | F1 | Acc | F1 | Acc | F1 | Acc | F1 |
| FOCAL-noPrivate | 0.939 | 0.938 | 0.8803 | 0.7229 | 0.8742 | 0.843 | 0.8146 | 0.8017 |
| FOCAL-noOrth | 0.9691 | 0.9688 | 0.9068 | 0.8218 | 0.9061 | 0.8967 | 0.828 | 0.7957 |
| FOCAL-wDistInd | 0.9223 | 0.9223 | 0.9493 | 0.8347 | 0.9438 | 0.9287 | 0.7921 | 0.7344 |
| FOCAL-noTemp | 0.9557 | 0.9551 | 0.9461 | 0.872 | 0.9319 | 0.9237 | 0.8414 | 0.8162 |
| FOCAL-wTempCon | 0.9564 | 0.956 | 0.9255 | 0.8124 | 0.9353 | 0.9141 | 0.8497 | 0.8131 |
| FOCAL | **0.9732** | **0.9729** | **0.9516** | **0.8580** | **0.9382** | **0.9290** | **0.8588** | **0.8463** |

**Analysis:** The complete ablation results on DeepSense encoder are presented in Table 16. Similar to our observations with SW-T encoder, all of the three components introduced in FOCAL (private space, orthogonality constraint, and temporal constraint) contribute positively to the downstream performance. However, we do find the orthogonality constraint and the temporal constraint play a more important role in the performance improvement with the DeepSense encoder than that with SW-T encoder on ACIDS, RealWorld-HAR, and PAMAP2 datasets. Besides, it is noticeable that distributional independence contributes positively to FOCAL on ACIDS and RealWorld-HAR datasets but contributes negatively to FOCAL on MOD and PAMAP2 datasets. We leave it as future work to investigate more into the role of distributional independence in factorizing the latent space within the multimodal contrastive learning paradigm.

# G  Limitations and Potential Extensions

**Assumption on Modality Synchronization:** We assume the signals simultaneously arrived at all sensory modalities such that the information at different modalities is synchronized. However, in some scenarios, different signals propagate at significantly different speeds. For instance, light travels much faster than sound. The shared modality embeddings can not be directly matched for the same samples without signal synchronizations between the modalities.

**Computational Complexity of Pretraining Loss:** In the current design, we take all pairs of modalities to compute their shared space consistency loss and private space orthogonality loss, which leads to $O(K^2)$ complexity to the number of modalities $K$. On one hand, we assume the modality number is limited to a handful count in most sensing applications; on the hand, we leave it as one of our future work to reduce the computational complexity in pretraining loss calculation.

**Dependency on Data Augmentations:** Our current contrastive learning paradigm is still not fully self-supervised, because we need to design a set of transformations (*i.e.*, data augmentations) for the private modality feature learning. However, different from image data, designing proper label-invariant data augmentations for time-series data can be challenging in some applications, especially when we do not have knowledge about the potential downstream tasks. One potential solution is to integrate the masked reconstruction learning paradigm into the framework, such that data augmentations can be avoided or less depended on.

**Multi-Device Collaboration:** This paper focused on multi-modal collaborative sensing settings while multi-device collaboration is not fully considered. The general design of contrastive learning in factorized latent space is extensible to the multi-device setting, but more designs need to be introduced to further address the heterogeneity contained in different vantage points and the scalability issues related to the number of participating sensor nodes in large-scale distributed sensing scenarios.

**Resiliency Against Domain Shift:** Although FOCAL improves the downstream performance of contrastive learning from multimodal sensing signals, it still exhibits relatively low accuracy in speed classification when data is collected from a different environment. There are multiple environmental factors that can lead to such degradations, including terrain, wind, sensor facing directions. We hope to integrate domain-invariant considerations into the learning objective in the future such that apparently task-unrelated information is decoupled and removed from the pretrained embedding space, and the model resiliency can be significantly enhanced.

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
