# OpenReview forum: "FOCAL: Contrastive Learning for Multimodal Time-Series Sensing Signals in Factorized Orthogonal Latent Space"
_NeurIPS.cc/2023/Conference — NeurIPS 2023 poster_

### Official Review · Reviewer_8mBK · 2023-07-01

**Soundness:** 3 good
**Presentation:** 3 good
**Contribution:** 3 good
**Rating:** 5
**Confidence:** 4

**Summary:**

This paper proposes a self-supervised contrastive learning framework for extracting comprehensive features from multimodal time-series sensing signals. The framework refines contrastive learning by modeling shared and private features and exploiting orthogonality constraints. Moreover, a Temporal Structural Constraint is proposed to make up for the locality of processing the temporal information locality appropriately in contrastive learning. Finally, to demonstrate the effectiveness of the proposed FOCAL, the authors conduct extensive experiments on multiple multimodal sensing datasets.

**Strengths:**

1. Extensive experiments are conducted on four benchmarks: MOD, ACIDS, RealWorld-HAR and PAMAP.
2. This work is relatively novel and technically sound.
3. The description and explanation of motivation are clear.
4. It seems to be a pioneer work to consider and solve temporal information locality.
5. The results are promising.

**Weaknesses:**

1. In the experiments, what is the intuition of the hyperparameter setting of Eq.5? Sensitivity analysis appears to be lacking.
2. Whether the proposed temporal structural constraint is also applicable to other comparative learning frameworks to solve the problem of temporal information locality.
3. From the ablation results, the private space seems to be important for contrastive learning, somewhat similar to intra-modal contrastive loss. This work is based on strong orthogonality constraints, but in fact, the performance of the FOCAL-noOrth setting has not dropped too much. I am worried whether the orthogonality assumption is too strong.
4. Eq.4 is essentially a triplet contrastive loss, are there possible alternatives? Moreover, the value of the “margin ” does not seem to be mentioned in the paper.
5. I think Figure 6 should be compared with the visualization results of the best baseline (or some variants of FOCAL), so as to reflect the differences or superiority. like Supervised vs. MoCo vs. FOCAL? Otherwise, it feels like visualization for visualization's sake.

**Questions:**

See Weaknesses.

**Limitations:**

The authors provide a comprehensive discussion about limitations and potential extensions in the appendix.

---

> ### Author Rebuttal · Authors · 2023-08-09
>
> # ****Response to Reviewer 8mBK****
>
> ********************Q1:******************** In the experiments, what is the intuition of the hyperparameter setting of Eq.5? Sensitivity analysis appears to be lacking.
>
> ****************Response****************: We have added the sensitivity test and plotted the performance of FOCAL against different hyperparameters in Figure 12 in the attached pdf:, and we observe that FOCAL is generally robust against the hyperparameter selections, with less than 2% accuracy fluctuations in all cases. For this reason, we did not perform a comprehensive hyperparameter search in our experiment.
>
> ****************Q2****************: Whether the proposed temporal structural constraint is also applicable to other comparative learning frameworks to solve the problem of temporal information locality.
>
> ****************Response****************: As in response to Reviewer zwVA, we applied the proposed temporal constraint to multiple contrastive learning baselines (i.e., SimCLR, MoCo, CMC, Cocoa, and GMC). Table 15 and Table 16 summarize the results on ACIDS and PAMAP2, and we observed up to 18.99% improvement on ACIDS and up to 8.39% improvement on PAMAP2 in accuracy. It validates that the temporal constraint can be used as a plugin to enhance existing contrastive learning frameworks for time-series data.
>
> **Table 15: Benefits of Temporal Constraints to SOTA baselines on ACIDS**
>
> |  | SimCLR |  | MoCo |  | CMC |  | Cocoa |  | GMC |  |
> | --- | --- | --- | --- | --- | --- | --- | --- | --- | --- | --- |
> |  | Acc | F1 | Acc | F1 | Acc | F1 | Acc | F1 | Acc | F1 |
> | wTemp | **0.7461** | **0.6938** | **0.7836** | **0.6618** | **0.8690** | 0.7090 | **0.8543** | **0.7665** | **0.9347** | **0.8109** |
> | Vanilla | 0.7438 | 0.6101 | 0.7717 | 0.6205 | 0.8443 | **0.7244** | 0.6644 | 0.5359 | 0.9096 | 0.7929 |
>
> **Table 16: Benefits of Temporal Constraints to SOTA baselines on PAMAP2**
>
> |  | SimCLR |  | MoCo |  | CMC |  | Cocoa |  | GMC |  |
> | --- | --- | --- | --- | --- | --- | --- | --- | --- | --- | --- |
> |  | Acc | F1 | Acc | F1 | Acc | F1 | Acc | F1 | Acc | F1 |
> | wTemp | **0.7129** | **0.6884** | **0.7800** | **0.7602** | 0.7804 | 0.7583 | **0.8442** | **0.8146** | **0.8253** | **0.8114** |
> | Vanilla | 0.6802 | 0.6583 | 0.7559 | 0.7387 | **0.7906** | **0.7706** | 0.7603 | 0.7187 | 0.8119 | 0.7860 |
>
>
>
> ****************Q3****************: This work is based on strong orthogonality constraints, but in fact, the performance of the FOCAL-noOrth setting has not dropped too much. I am worried whether the orthogonality assumption is too strong.
>
> ****************Response****************: Our interpretation is the introduction of the private space brings the most performance improvement (FOCAL vs. FOCAL-noPrivate), while the orthogonality constraint further enhances the performance by pushing the two spaces to exploit unrelated semantics. As can be seen from our ablation study, FOCAL-noOrth improves significantly over FOCAL-noPrivate, while FOCAL further improves over FOCAL-noOrth, which means the orthogonality constraint contributes positively to FOCAL. Alternatively, when we replace the geometrical orthogonality with the distributional independence, FOCAL-wDistInd degrades significantly compared to FOCAL-noOrth, which means the distributional independence contributes negatively to the framework.
>
> ****Q4****: Eq.4 is essentially a triplet contrastive loss, are there possible alternatives? Moreover, the value of the “margin ” does not seem to be mentioned in the paper.
>
> ****************Response****************: We conducted a sensitivity test on the margin value in the temporal triplet loss and plotted the performance of FOCAL against different margin values in Figure 12(d) in the attached pdf. FOCAL's accuracy changes from 95.11% to 94.89% on ACIDS and from 82.22% to 84.22% on PAMAP2, when the margin varies from 0.1 to 3. Setting the margin as 1 is overall the best strategy. We also conclude that FOCAL is not very sensitive to the temporal constraint margin selection.
>
> ****************Q5****************: Figure 6 should be compared with the visualization results of the best baseline (or some variants of FOCAL), so as to reflect the differences or superiority. like Supervised vs. MoCo vs. FOCAL?
>
> ****************Response****************: We provided the t-SNE visualization comparison between FOCAL and several multi-modal contrastive learning frameworks (i.e., CMC, GMC, and Cocoa). Figure 10 and Figure 11 in the attached pdf visualize their concatenated modality features with DeepSense and SW-T encoder, respectively, on the MOD dataset as an example. The results show that FOCAL achieved better separation among different classes, which is aligned with our quantitative clustering evaluation results.

---

> ### Author Response · Authors · 2023-08-19
>
> Dear Reviewer 8mBK,
>
> Thanks again for your valuable feedback.
>
> We have carefully considered all your comments and have tried our best to improve the paper. We have added a sensitivity test for each hyper-parameter showing the robustness of FOCAL against these hyper-parameters, t-SNE figures demonstrating a better separation among different classes for FOCAL, and additional experiments with improvements up to 18.99% after applying temporal constraints to multiple contrastive learning baselines.
>
> As we are reaching the end of the author-reviewer discussion period, we wonder if our responses have sufficiently addressed your concerns, and we are happy to address any questions you may have.
>
> Thanks again for your effort in helping us improve the paper!
>
> Best,
>
> Submission #3273 Authors

---

> > ### Comment · Reviewer_8mBK · 2023-08-19
> >
> > Thank you for your kind response. The authors have addressed some of my concerns. I'd like to raise my rating.

---

### Official Review · Reviewer_Q2hY · 2023-07-04

**Soundness:** 2 fair
**Presentation:** 2 fair
**Contribution:** 2 fair
**Rating:** 5
**Confidence:** 5

**Summary:**

This paper proposes a multimodal contrastive learning framework (FOCAL) for extracting comprehensive features from multimodal time-series sensing signals through self-supervised training. In which, FOCAL first decouples each modality into two subspaces, i.e., shared and private spaces, and uses simple soft orthogonal loss as an objective. Then, the temporal structural constraint for modality features is proposed to enhance feature temporally. The authors use four multimodal sensing datasets with two backbone encoders and two classifiers to conduct many experiments.

**Strengths:**

1. The motivation and contribution of the paper are clearly described.
2. Many experiments are presented in the manuscript.

**Weaknesses:**

1.	The biggest concern is the effectiveness of multimodal feature factorization. From the line 161, this work uses the simple soft orthogonal constraint to decouple or factorize each modality feature. However, the single orthogonal constraint is too simple and may not decouple multimodal features well. In particular, the information is easily leaked between modalities, and many prior works have tried to solve this problem such as [1-3]. In addition, there is no visualization result for both subspaces in the experimental part, so it is difficult to determine whether the factorization is effectiveness.

[1] Li Y, Wang Y, Cui Z. Decoupled Multimodal Distilling for Emotion Recognition[C]//Proceedings of the IEEE/CVF Conference on Computer Vision and Pattern Recognition. 2023: 6631-6640.

[2] Ouyang J, Adeli E, Pohl K M, et al. Representation disentanglement for multi-modal brain MRI analysis[C]//Information Processing in Medical Imaging: 27th International Conference, IPMI 2021, Virtual Event, June 28–June 30, 2021, Proceedings 27. Springer International Publishing, 2021: 321-333.

[3] Yang D, Huang S, Kuang H, et al. Disentangled representation learning for multimodal emotion recognition[C]//Proceedings of the 30th ACM International Conference on Multimedia. 2022: 1642-1651.

2.	In the line 29, the word “ignore” may not be as appropriate. For example, the CLIP contrastive frameworks does not emphasize shared or private spaces, but we cannot assume that this work ignores heterogeneity, only that they do not explicitly consider. Therefore, it might be better to replace "ignore" with "explicitly consider".
3.	In the experiments, why not consider some more common multimodal scenarios, such as the MOSI [4] and MOSEI [5] datasets, which contain the three most common modalities in the real world, i.e., Language, Visual and Acoustics, and whose features are packaged as time-series-based, in line with the settings in this paper.

[4] Zadeh A, Zellers R, Pincus E, et al. Mosi: multimodal corpus of sentiment intensity and subjectivity analysis in online opinion videos[J]. arXiv preprint arXiv:1606.06259, 2016.

[5] Zadeh A A B, Liang P P, Poria S, et al. Multimodal language analysis in the wild: Cmu-mosei dataset and interpretable dynamic fusion graph[C]//Proceedings of the 56th Annual Meeting of the Association for Computational Linguistics (Volume 1: Long Papers). 2018: 2236-2246.

4.	The public datasets lack the necessary references, please check carefully. (Line 199-206)

**Questions:**

Will the MOD dataset be open-released?

**Limitations:**

No limitations are provided in this paper.

---

> ### Author Rebuttal · Authors · 2023-08-09
>
> # **Response to Reviewer Q2hY**
>
> **Q1**: The biggest concern is the effectiveness of multimodal feature factorization. From line 161, this work uses the simple soft orthogonal constraint to decouple or factorize each modality feature. However, the single orthogonal constraint is too simple and may not decouple multimodal features well. In particular, information is easily leaked between modalities, and many prior works have tried to solve this problem such as [1-3]. In addition, there is no visualization result for both subspaces in the experimental part, so it is difficult to determine whether the factorization is effective.
>
> **Response** :
>
> We agree with the reviewer that FOCAL shares with reference [1-3] in exploiting both shared information and private information in multi-modal collaboration. However, there are several factors that make FOCAL substantially different from the listed references.
> First, regarding the learning paradigm, [1, 2, 3] all work with supervised tasks (classification or reconstruction) where the disentanglement objectives are used as augmentations to original task objectives. However, FOCAL works with self-supervised contrastive learning, where the factorization space should be designed without knowledge of downstream tasks and the auxiliary objectives should cooperate with the contrastive learning objectives, which are challenges that have not been considered in listed references.
> Second, regarding the application scenario, we have positioned the paper as a self-supervised learning framework for multi-model time-series sensing signals where contributions are made from both the multi-modal perspective and time-series perspective. While we think part of the design could be potentially applicable to more diverse application scenarios, the evaluation of language-vision-acoustics task is beyond the scope of this paper and we leave it as our future explorations.
>
> We select the simple orthogonal constraint according to the intuition that it aligns well with the contrastive learning paradigm and the fact that it benefits the downstream task performance.
> On the one hand, contrastive learning works in a manner that the angular similarity between embeddings represents semantical proximity, while the geometrical orthogonal constraint is aligned with semantical independence in contrastive learning and penalizes the overlaps between the private space and shared space.
> On the other hand, adding the orthogonal constraint benefits the downstream tasks after the contrastive pretraining phase. As we presented in the ablation study, although the alternative distributional independence might create better-disentangled representations, it leads to significant degradation after being integrated with the contrastive objectives.
>
> As for the information leakage concern (i.e., exactly the same information is exploited in both spaces), we believe the proposed orthogonal constraint is a valid solution that prevents leakage while being suitable for contrastive learning, as demonstrated by the end results in our ablation study. The performance improvement of FOCAL-noOrth to FOCAL-noPrivate means the private space task encourages the encoder to exploit more comprehensive semantics, while the improvement of FOCAL to FOCAL-noOrth further validates that the orthogonal constraint encourages the private space to exploit non-overlapped information as the shared space and enriches the learned semantics. We do not expect the private modality representations to be fully separable in the visualization, because the instance discrimination task selected for the private space is not only related to private modality information but can also be related to the shared information across modalities. However, too much reliance on the shared information in the private space is penalized by the posed orthogonal constraints. How to design a unique proxy task that is only related to the private modality information is a challenging task that we want to investigate in the future. The margin loss solution in [1] can not be applied because we do not have downstream category information during the self-supervised pretraining.
>
> **Q2**: In line 29, the word “ignore” may not be as appropriate.
>
> **Response**: We agree with the reviewer and will change “ignore” to “explicitly consider” in our next draft.
>
> **Q3**: In the experiments, why not consider some more common multimodal scenarios, such as the MOSI [4] and MOSEI [5] datasets, which contain the three most common modalities in the real world, i.e., Language, Visual, and Acoustics, and whose features are packaged as time-series-based, in line with the settings in this paper.
>
> **Response**: Thanks for the suggestion. Due to the short duration of the rebuttal period, we were not able to extend our framework to more diverse modalities (i.e., language, vision, and acoustics). We would leave this effort as one of our potential future extensions. Instead, we position FOCAL as a novel contrastive learning framework for multi-model time-series sensing signals in IoT applications and compare it with an extensive set of SOTA baselines for the same setting.
>
> **Q4**: The public datasets lack the necessary references, please check carefully. (Line 199-206)
>
> **Response**: Thanks for pointing out the issue. Although the references are added in the Appendix, we will also add the missing references in the main body of the paper.
>
> **Q5**: Will the MOD dataset be open-released?
>
> **Response**: Yes, we do plan to release the MOD dataset upon the paper's acceptance.
>
> **Q6** No limitations are provided in this paper.
>
> **Response**: We have put the discussion on limitations and future extensions in Appendix G due to the space limit. We will also integrate the comments from the reviewers into more limitation discussions in the next version.

---

> > ### Comment · Reviewer_Q2hY · 2023-08-22
> >
> > I have read the response carefully, and my question can be addressed basically. I would like to increase my score.

---

> ### Author Response · Authors · 2023-08-19
>
> Dear Reviewer Q2hY,
>
> Thanks again for your valuable feedback.
>
> We have explained that while FOCAL shares similarities with reference [1-3], several factors make FOCAL substantially different from the listed reference. We then clarified the motivation and intuition of choosing the proposed orthogonal constraint over a few alternatives. We further supported our claims through ablation studies that demonstrate the effectiveness of the orthogonal constraints and their compatibility with contrastive learning. We believe that the proposed orthogonal constraint is a valid solution to address the information leakage for multimodal contrastive learning. From the ablation studies, we observed an improvement for FOCAL compared to FOCAL-noOrth, validating that the orthogonal constraint encourages the private space to exploit non-overlapped information as the shared space and enriches the learned semantics.
> We have also fixed the wording and promise to release the MOD dataset upon the paper’s acceptance.
> Lastly, we would like to kindly note that we position FOCAL as a novel self-supervised learning framework in the multimodal time-series sensing domain. Although we believe some aspects could be potentially applicable to more diverse applications, the evaluation of the language-vision-acoustics task is beyond the scope of this paper, and we leave it as our future explorations.
>
> As we are reaching the end of the author-reviewer discussion period, we wonder if our responses have sufficiently addressed your concerns, and we are happy to address any questions you may have.
>
> Thanks again for your effort in helping us improve the paper!
>
> Best,
>
> Submission #3273 Authors

---

### Official Review · Reviewer_oGNZ · 2023-07-07

**Soundness:** 3 good
**Presentation:** 3 good
**Contribution:** 3 good
**Rating:** 6
**Confidence:** 3

**Summary:**

The paper proposes FOCAL, a contrastive learning method for multimodal time-series signals. The main idea is to first encode each modality into a factorized orthogonal space, and design four pretraining objectives that enforce modality consistency, transformation consistency, orthogonality constraint and temporal locality constraint. Experimental results demonstrate the efficacy of FOCAL.

**Strengths:**

+ The two motivations for designing the contrastive objectives make sense to me --- modality-private features can contribute in contrastive learning, and the temporal structural constraint also seems valid.

+ The authors conduct an extensive evaluation of FOCAL --- it is compared with many contrastive learning baselines and evaluated on four multimodal datasets. The improvement achieved by FOCAL is large.

+ The ablation study is great and provides a good analysis on the four pretraining objectives.

+ The paper is clearly presented and easy to follow.

**Weaknesses:**

While I generally agree with the motivations and the proposed four objectives, Eq (5) is a combination of four objectives, and it seems hard to balance the four terms. I wonder if some of the training objectives may compete during training, and how do the authors set the three hyper-parameters $\lambda_p$, $\lambda_o$ and $\lambda_t$? Does it require some manual hyper-parameter tuning? Jointly optimizing the four objectives does not look like an easy task.

**Questions:**

See weaknesses.

**Limitations:**

Yes the limitations have been well discussed in Appendix.

---

> ### Author Rebuttal · Authors · 2023-08-09
>
> # ****Response to Reviewer oGNZ****
>
> ****************Q1****************: I wonder if some of the training objectives may compete during training, and how do the authors set the three hyper-parameters $\lambda_p$, $\lambda_o$, and $\lambda_t$? Does it require some manual hyperparameter tuning? Jointly optimizing the four objectives does not look like an easy task.
>
> **Response**: Our conclusion is that competition between learning objectives does not happen in FOCAL, for the following reasons. First, without the private space task, the performance of FOCAL-noPrivate is much lower than FOCAL-noOrth (note noPrivate also implicitly means no orthogonality constraint), thus the private contrastive loss contributes positively to FOCAL. Second, for the orthogonality constraint and temporal constraint, FOCAL works better than FOCAL-noOrth and FOCAL-noTemp variants respectively, thus our orthogonal constraint and temporal constraint both contribute positively. Instead, we observed that FOCAL-wDistInd works worse than FOCAL-noOrth, while FOCAL-wTempCon works worse than FOCAL-noTemp, which means the competition happens between these alternative constraints and the original contrastive objectives. As for the hyper-parameters, they are mostly shared across different datasets and backbone encoders. We set $\lambda_p$ as 1 and $\lambda_o$ as 3 by default and only tune $\lambda_t$ manually when needed. As mentioned in our general response, we have added the sensitivity test results in the attached pdf to demonstrate the resiliency of FOCAL against these hyperparameter values.

---

> > ### Comment · Reviewer_oGNZ · 2023-08-17
> > **Thanks for the rebuttal!**
> >
> > The provided responses have addressed my concerns.

---

### Official Review · Reviewer_WTA9 · 2023-07-08

**Soundness:** 3 good
**Presentation:** 4 excellent
**Contribution:** 3 good
**Rating:** 7
**Confidence:** 2

**Summary:**

The paper introduces FOCAL, a new contrastive learning framework for extracting comprehensive features from multimodal time-series sensing signals through self-supervised training. Unlike previous frameworks that focused solely on shared information between sensory modalities, FOCAL also factors in exclusive modality information crucial for understanding physical semantics. It also properly handles temporal information locality, which previous time series contrastive frameworks have not done. FOCAL creates a factorized latent space of shared and private (modality-exclusive) features from each modality and applies a temporal structural constraint on these features. Through extensive testing on four multimodal sensing datasets, FOCAL consistently outperforms the state-of-the-art baselines in downstream tasks, demonstrating its superior performance.

**Strengths:**

- This paper is excellently written, with clear logic and motivation that is easy to understand.
- The results presented are impressive and Figure 1 does a fantastic job in aiding the understanding of the paper.
-  The idea of modeling modality-shared and specific information is very great.
Great work!




**Weaknesses:**

I couldn't find any significant weaknesses in this paper. However, I should note that I'm not an expert in this topic. While I am familiar with multimodality, and contrastive, my understanding of time series analysis is limited

**Questions:**

- Is there any hyperparameter search performed for λp, λo, and λt which control the weights of each loss component? Additionally, was there any hyperparameter search carried out for other methods?"
- "Why does the FOCAL-wDistInd perform worse than the noOrth model? Does the replacement of orthogonality with distributional independence contribute to this difference?

---

> ### Author Rebuttal · Authors · 2023-08-09
>
> # ****Response to Reviewer WTA9****
>
> ****************Q1****************: Is there any hyperparameter search performed for $\lambda_p$, $\lambda_o$, and $\lambda_t$ which control the weights of each loss component? Additionally, was there any hyperparameter search carried out for other methods?
>
> ****************Response****************: We did not perform a comprehensive hyperparameter search for the loss weights in FOCAL, which we agree might slightly improve our performance. Instead, we manually tune the hyperparameters on one dataset and use the same hyperparameters across the remaining datasets, with minor changes (i.e., on $\lambda_t$). The same tuning strategy is also carried out for all baseline methods. As mentioned in our general response, we have added the sensitivity test results in Figure 12 in the attached pdf to demonstrate the resiliency of FOCAL against these hyperparameter values.
>
> ****************Q2****************: Why does the FOCAL-wDistInd perform worse than the noOrth model? Does the replacement of orthogonality with distributional independence contribute to this difference?
>
> ****Response****: As we observed in the experiment, the distributional independence constraint leads to significant performance degradation in most cases, even worse than the FOCAL-noOrth, especially when SwinTransformer is used as the backbone encoder. Our interpretation is two-fold: First, the dynamic nature of the attention mechanism within SwinTransformer makes the distribution of encoded latent features more complicated and hard to discriminate with a simple classifier; Second, the iterative training between the distribution discriminator and the contrastive objectives does not work coherently as with the autoencoder training. It was easy to collapse in our experiments. We leave the effective integration of distributional disentanglement and contrastive learning framework as a future exploration direction.

---

> > ### Comment · Reviewer_WTA9 · 2023-08-13
> > **Thanks for reply**
> >
> > I don't have other questions.

---

### Official Review · Reviewer_zwVA · 2023-07-27

**Soundness:** 3 good
**Presentation:** 3 good
**Contribution:** 3 good
**Rating:** 7
**Confidence:** 3

**Summary:**

The proposed FOCAL is a constrasitive learning method or time-series signals. The major contributions are:
1. For multi-modality samples, FOCAL learns shared feature that are similar between modalities, and a private features that are similar intra-modality but different across modalities.
2. The shared and private features are orthoigonal, so that they focus on different aspects.
3. The Temporal Locality Constraint which restricts the average distance within a short sequence to be less than the average distance between two random sequences. The "average" operation allows some samples to be similar even though they are from distant samples in time-domain.

Extensive experiments are conducted with 4 datasets and 12 baselines, which gives consistent SOTA performance on classification and clustering tasks.

**Strengths:**

1. The idea of private feature is novel, and effective according to the ablation study.
2. Although the temporal structural constraint is a simple modification, it provides noticable performance increase.
3. The proposed method is simple and easy to follow. Detailed network design, data processing, training configurations are provided in Appendix.

**Weaknesses:**

1. The proposed contribution is mostly the loss functions, e.g., private and shared feature constrative loss, temporal locality loss, etc. I believe these loss can be applied to existing methods with simple modification. It would be very strong to show that the proposed loss serves as a plugin that enhance other contrastive learning methods.

**Questions:**

See weakness.

**Limitations:**

The paper discussed little on the techinical limitation and future directions. It would be great the discuss more on failure cases.

---

> ### Author Rebuttal · Authors · 2023-08-09
>
> # ****Response to Reviewer zwVA****
>
> **Q 1**: It would be very strong to show that the proposed loss serves as a plugin that enhances other contrastive learning methods.
>
> **Response**: Thanks for the suggestion. We have applied the proposed temporal constraint to multiple contrastive learning baselines (i.e., SimCLR, MoCo, CMC, Cocoa, and GMC). Table 15 and 16 summarize the results on ACIDS and PAMAP2, and we have observed noticeable performance improvement in most cases (up to 18.99% on ACIDS and up to 8.39% on PAMAP2). It validates that the temporal constraint can be used as a plugin to enhance existing contrastive learning frameworks for time-series data.
>
> **Q 2**: The paper discussed little on the technical limitation and future directions. It would be great the discuss more on failure cases.
>
> **Response**: We would like to note that the limitations and potential extensions of this paper are discussed in Appendix G. We will also expand this list by integrating the comments made by all reviewers in the next version.
>
>
> **Table 15: Benefits of Temporal Constraints to SOTA baselines on ACIDS**
>
> |  | SimCLR |  | MoCo |  | CMC |  | Cocoa |  | GMC |  |
> | --- | --- | --- | --- | --- | --- | --- | --- | --- | --- | --- |
> |  | Acc | F1 | Acc | F1 | Acc | F1 | Acc | F1 | Acc | F1 |
> | wTemp | **0.7461** | **0.6938** | **0.7836** | **0.6618** | **0.8690** | 0.7090 | **0.8543** | **0.7665** | **0.9347** | **0.8109** |
> | Vanilla | 0.7438 | 0.6101 | 0.7717 | 0.6205 | 0.8443 | **0.7244** | 0.6644 | 0.5359 | 0.9096 | 0.7929 |
>
> **Table 16: Benefits of Temporal Constraints to SOTA baselines on PAMAP2**
>
> |  | SimCLR |  | MoCo |  | CMC |  | Cocoa |  | GMC |  |
> | --- | --- | --- | --- | --- | --- | --- | --- | --- | --- | --- |
> |  | Acc | F1 | Acc | F1 | Acc | F1 | Acc | F1 | Acc | F1 |
> | wTemp | **0.7129** | **0.6884** | **0.7800** | **0.7602** | 0.7804 | 0.7583 | **0.8442** | **0.8146** | **0.8253** | **0.8114** |
> | Vanilla | 0.6802 | 0.6583 | 0.7559 | 0.7387 | **0.7906** | **0.7706** | 0.7603 | 0.7187 | 0.8119 | 0.7860 |

---

> > ### Comment · Reviewer_zwVA · 2023-08-11
> > **Additional experiment on private and shared feature constrative loss**
> >
> > The benefit of temporal constraint is proven by Table 15 and 16. However, the major contribution of the paper is the shared and private loss, and the Orthogonality loss. I would suggest similar experiments to be done, i.e., enhance baselines with the proposed private and shared feature constrative loss, Orthogonality loss.

---

> > > ### Author Response · Authors · 2023-08-13
> > > **Additional results on benefits of private and shared feature contrastive loss**
> > >
> > > Thank you for the suggestion! We have conducted additional experiments to assess the performance of selected baselines with our proposed factorized contrastive and orthogonal loss. The tables below summarize the results on ACIDS and PAMAP2 datasets using DeepSense as the backbone model.  For these experiments, we would like to preserve the original design of baseline approaches while integrating our proposed loss objectives. Most baselines either target primarily on instance discrimination (e.g., SimCLR, MoCo) or have designs that do not align well with subspace factorizations (e.g., Cocoa, Cosmo). We eventually selected CMC, GMC, and TS-TCC as the main subjects of the experiments.
> > >
> > > For CMC, as introduced in our submission, the CMC loss is applied to the shared modality embeddings, while the instance discrimination loss is applied to each private modality space. Besides, orthogonality constraints are applied between shared-private and private-private modality embeddings.
> > >
> > > For GMC, we randomly generate two augmented views. For each view, we apply GMC’s original loss objective to the **shared space embeddings** across the modalities. Then we measure the NT-Xent loss on the private embeddings of the two views for private loss. Lastly, we added the orthogonal loss between the factorized subspaces.
> > >
> > > FOR TS-TCC, we consider TS-TCC’s original loss objective as the **private loss** since it aims to learn temporal representation within a single modality. To measure the shared loss, we calculate the NT-Xent loss by contrasting the shared space embedding of each modality pair. Lastly, we added the orthogonal loss between the factorized subspaces.
> > >
> > > We can see from the tables that introducing our proposed loss methods has improved the performance of the baselines (relatively) by **up to 12.00% in ACIDS and 5.83% in PAMAP2**. This demonstrates the effectiveness of the factorized contrastive orthogonal loss as an enhancement to the existing contrastive learning frameworks.
> > >
> > > Please let us know if you have any further concerns or comments.
> > >
> > > **Table 17: Benefits of Factorized Contrastive Orthogonal Constraints to SOTA baselines on ACIDS**
> > >
> > > |  | CMC |  | GMC |  | TS-TCC |  |
> > > | --- | --- | --- | --- | --- | --- | --- |
> > > |  | Acc | F1 | Acc | F1 | Acc | F1 |
> > > | wOrth | **0.9456** | **0.8014** | **0.9343** | **0.8174** | **0.8032** | **0.7093** |
> > > | Vanilla | 0.8443 | 0.7244 | 0.9096 | 0.7929 | 0.7667 | 0.6164 |
> > >
> > > **Table 18: Benefits of Factorized Contrastive Orthogonal Constraints to SOTA baselines on PAMAP2**
> > >
> > > |  | CMC |  | GMC |  | TS-TCC |  |
> > > | --- | --- | --- | --- | --- | --- | --- |
> > > |  | Acc | F1 | Acc | F1 | Acc | F1 |
> > > | wOrth | **0.8367** | **0.8255** | **0.8166** | **0.7892** | **0.7863** | **0.7484** |
> > > | Vanilla | 0.7906 | 0.7706 | 0.8119 | 0.7860 | 0.7772 | 0.7246 |

---

### Author Rebuttal · Authors · 2023-08-09

# General Responses

We would like to sincerely thank all the reviewers for their valuable feedback and constructive suggestions for this submission. As a summary of our responses, we have finished the following tasks during the rebuttal period:

1. We added a sensitivity test of loss hyperparameters on ACIDS and PAMAP2 datasets, including three loss weight terms ($\lambda_p$, $\lambda_o$, $\lambda_t$) and the temporal constraint margin. We present the accuracy of FOCAL with different hyperparameters in Figure 12, and the results show that FOCAL is generally resilient against these hyperparameters. Therefore, the hyperparameter values are mostly shared across different datasets and backbone encoders, with manual tuning only on temporal constraint weight $\lambda_t$. In addition, we set the private loss weight $\lambda_p$ as $1$, the orthogonal loss weight $\lambda_o$ as $3$, and the temporal loss margin as $1$ by default.
2. As suggested by Reviewer zwVA and Reviewer 8mBK, we applied the proposed temporal constraint to multiple baselines (i.e., SimCLR, MoCo, CMC, Cocoa, and GMC), and achieved noticeable performance improvement compared to their vanilla versions. The full results on ACIDS and PAMAP2 datasets are summarized in Table 15 and Table 16. The inspiring results demonstrated the effectiveness of the proposed temporal constraint in general contrastive learning frameworks for time-series data.
3. We provided the visualization comparison between FOCAL and several multi-modal contrastive baselines in Figure 10 and Figure 11, which showed that FOCAL achieved better separation among different downstream classes after the pretraining.
4. Due to the space limit, we have put the discussion of technical limitations and potential extensions in Appendix G, and we will accordingly expand this list by integrating the comments of all reviewers.
5. We promise to fix all presentation and terminology issues pointed out by Reviewer Q2hY.

**We have included these tables (Table 16-17) and figures (Figure 10-12) in the attached pdf to address all key concerns and to clarify our work.**

---

### Author Response · Authors · 2023-08-17
**Friendly Reminder: Author-Reviewer discussion phase**

Dear Reviewers,

We hope this message finds you well. As we approach the final days of the author-reviewer discussion phase, we would like to kindly remind the reviewers to take a moment to review the rebuttals and provide any further feedback you might have. We genuinely appreciate your comments, and should you have any questions or concerns, please don't hesitate to let us know.

Best,

Submission #3273 Authors

---

### Decision · Program_Chairs · 2023-09-21

**Decision:**

Accept (poster)

**Comment:**

This paper proposes a new contrastive learning method for time-series. The reviewers noted the paper's strong experimental results and clarity in their approach. The rebuttal further convinced the reviewers and they recommended acceptance (5,6,5,7,7).